# Negative-prompt Inversion: Fast Image Inversion for Editing with Text-guided Diffusion Models

## Abstract

In image editing employing diffusion models, it is crucial to preserve the reconstruction quality of the original image while changing its style. Although existing methods ensure reconstruction quality through optimization, a drawback of these is the significant amount of time required for optimization. In this paper, we propose **negative-prompt inversion**, a method capable of achieving equivalent reconstruction solely through forward propagation without optimization, thereby enabling much faster editing processes. We experimentally demonstrate that the reconstruction quality of our method is comparable to that of existing methods, allowing for inversion at a resolution of 512 pixels and with 50 sampling steps within approximately 5 seconds, which is more than 30 times faster than null-text inversion. Reduction of the computation time by the proposed method further allows us to use a larger number of sampling steps in diffusion models to improve the reconstruction quality with a moderate increase in computation time.

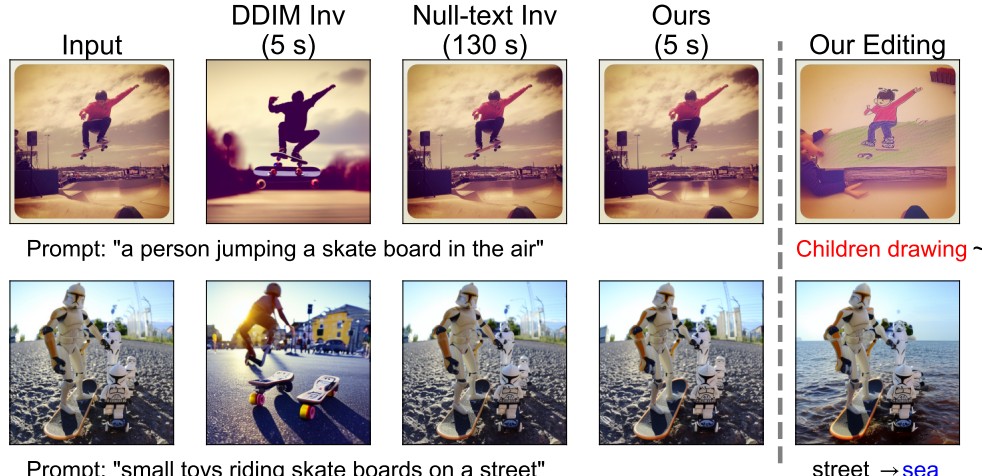

Figure 1: **Negative-prompt inversion.** Comparison in reconstruction quality and time between the proposed method (negative-prompt inversion; Ours), DDIM inversion (Song et al., 2021a; Dhariwal & Nichol, 2021), and null-text inversion (Mokady et al., 2023). The rightmost column shows the results of image editing obtained using prompt-to-prompt (Hertz et al., 2023) with our reconstruction.

## 1 Introduction

Diffusion models (Ho et al., 2020) are known to yield high-quality results in the fields of image generation (Song & Ermon, 2019; Ho et al., 2020; Song et al., 2021b; Saharia

et al., 2022; Dhariwal & Nichol, 2021; Rombach et al., 2022), video generation (Harvey et al., 2022; Ho et al., 2022; Höppe et al., 2022; Blattmann et al., 2023), and text-to-speech conversion (Chen et al., 2021a;b). Text-guided diffusion models (Kim et al., 2022) are diffusion models conditional on given texts ("prompts"), which can generate data with various modalities that fit well with the prompts. It is known that by strengthening the text conditioning through classifier-guidance (Dhariwal & Nichol, 2021) or classifier-free guidance (CFG) (Ho & Salimans, 2021), the fidelity to the text can be improved further. In image editing using text-guided diffusion models, elements in images, such as objects and styles, can be changed with high quality and diversity guided by text prompts.

In applications based on image editing methods, one must be able to generate images that are of high fidelity to original images in the first place, including reproduction of their details, and then one will be able to perform appropriate editing of images according to the prompts therefrom. To achieve high-fidelity image generation, most existing research exploits optimization of parameters such as model weights, text embeddings, and latent variables, which results in high computational costs and memory usage.

In this paper, we propose a method that can obtain latent variables and text embeddings yielding high-quality reconstruction of real images while using only forward computations. Our method requires neither optimization nor backpropagation, enabling faster processing and reducing memory usage. The proposed method is based on null-text inversion (Mokady et al., 2023), which has the denoising diffusion implicit model (DDIM) inversion (Song et al., 2021a; Dhariwal & Nichol, 2021) and CFG as its principal building blocks. Null-text inversion improves the reconstruction accuracy by optimizing an embedding which is used in CFG so that the diffusion process calculated by DDIM inversion aligns with the reverse diffusion process calculated using CFG. We discovered that the optimal solution embedding obtained by this method can be approximated by the embedding of the conditioning text prompt, and that editing also works by using an embedding of a source prompt instead of the optimal embedding.

Figure 1 shows a comparison between the proposed method and existing ones. Our method generated high-quality reconstructions when a real image and a corresponding prompt were given. DDIM inversion had noticeably lower reconstruction accuracy. Null-text inversion achieved high-quality results, nearly indistinguishable from the input image, but required more computation time. The proposed method, which we call **negative-prompt inversion**, allows for computation at the same speed as DDIM inversion, while achieving accuracy comparable to null-text inversion. Furthermore, combining our method with image editing methods such as prompt-to-prompt (Hertz et al., 2023) allows fast single-image editing (Editing).

We summarize our contributions as follows:

1. We propose a method for fast reconstruction of real images with diffusion models without optimization.

2. We experimentally demonstrated that our method achieves visually equivalent reconstruction quality to existing methods while enabling a more than 30-fold increase in processing speed.

3. Combining our method with existing image editing methods like prompt-to-prompt allows fast real image editing.

## 2 Related work

**Image editing by diffusion models.** In the field of image editing using diffusion models such as Imagen (Saharia et al., 2022) and Stable Diffusion (Rombach et al., 2022), Imagic (Kawar et al., 2023), UniTune (Valevski et al., 2023), and SINE (Zhang et al., 2023) are models for editing compositional structures, as well as states and styles of objects, in a single image. These methods ensure fidelity to original images via fine-tuning models and/or text embeddings.

Prompt-to-prompt (Hertz et al., 2023), another image editing methods based on diffusion models, reconstructs original images via making use of null-text inversion. Null-text inversion successfully reconstructs real images by optimizing the null-text embedding (the embedding for unconditional prediction) at each prediction step. All these methods attempt to reconstruct real images by incorporating an optimization process, which typically takes several minutes for editing a single image.

Plug-and-Play (Tumanyan et al., 2023) edits a single image without optimization. It obtains latent variables corresponding to the input image using DDIM inversion and reconstructs it according to the edited prompt, inserting attention and feature maps to preserve image structures. Our inversion method is independent of editing methods, allowing for the freedom to choose an editing method while maintaining a high-quality image structure regardless of the editing method.

**Image reconstruction by diffusion models.** Textual Inversion (Gal et al., 2023a) and DreamBooth (Ruiz et al., 2023) are methods that reconstruct common concepts from a few real images by fine-tuning the model. On the other hand, ELITE (Wei et al., 2023) and Encoder for Tuning (E4T) (Gal et al., 2023b) seek text embeddings that reconstruct real images using an encoder. The former ones are aimed at concept acquisition, making them difficult to reconstruct the original image with high fidelity. Although the latter ones require less computation time compared with the former ones, the ease of editing operations is limited, as the corresponding text is not explicitly obtained.

The proposed method realizes nearly the same reconstruction as null-text inversion, but with only forward computation, enabling image editing in just a few seconds. By combining our method with image editing methods such as prompt-to-prompt, it becomes possible to achieve flexible and advanced editing using text prompts.

## 3 METHOD

### 3.1 OVERVIEW

In this section, we describe our method for obtaining latent variables and text embeddings which reconstruct a real image using diffusion models without optimization. Our goal is that when given a real image $I$ and an appropriate prompt $P$, we calculate latent variables $(z_t)$, where $t$ is the index for the diffusion steps, in the reverse diffusion process so as to reconstruct $I$.

### 3.2 DDIM INVERSION

A diffusion model has a forward diffusion process over diffusion steps from 0 to $T$ (e.g., $T = 1000$ in Ho et al. (2020)), which degrades the representation $z_0$ of an original sample into a pure noise $z_T$, and an associated reverse diffusion process, which generates $z_0$ from $z_T$. In the training process, a latent variable $z_t$ for $t \in \{1, \cdots, T\}$ is calculated by adding noise $\epsilon$, and the model is trained to predict the noise $\epsilon$ from the latent variables $z_t$. In text-guided diffusion models, the model is further conditioned by an embedding $C$ of a text prompt $P$, which is obtained via a text encoder like CLIP (Radford et al., 2021). The loss function is the mean squared error (MSE) between the predicted noise $\epsilon_\theta$ and the actual noise $\epsilon$,

$$L(\theta) = \mathbb{E}_{t \sim U(1,T), \epsilon \sim \mathcal{N}(\mathbf{0},\mathbf{I})} \|\epsilon - \epsilon_\theta(z_t, t, C)\|_2^2,$$

where $U(1,T)$ denotes the uniform distribution on the set $\{1, \cdots, T\}$, and where $\mathcal{N}(\boldsymbol{\mu}, \boldsymbol{\Sigma})$ denotes the multivariate Gaussian distribution with mean $\boldsymbol{\mu}$ and covariance $\boldsymbol{\Sigma}$.

Stable Diffusion (Rombach et al., 2022) considers diffusion processes in a latent space: during the training process, a feature representation $z_0$ is obtained by passing a sample $x_0$ through an encoder. In the inference stage, a sample $x_0$ is generated by passing the generated representation $z_0$ through a decoder.

CFG is used to strengthen text conditioning. During the computation of the reverse diffusion process, the null-text embedding $\varnothing$, which corresponds to the embedding of a null text "",

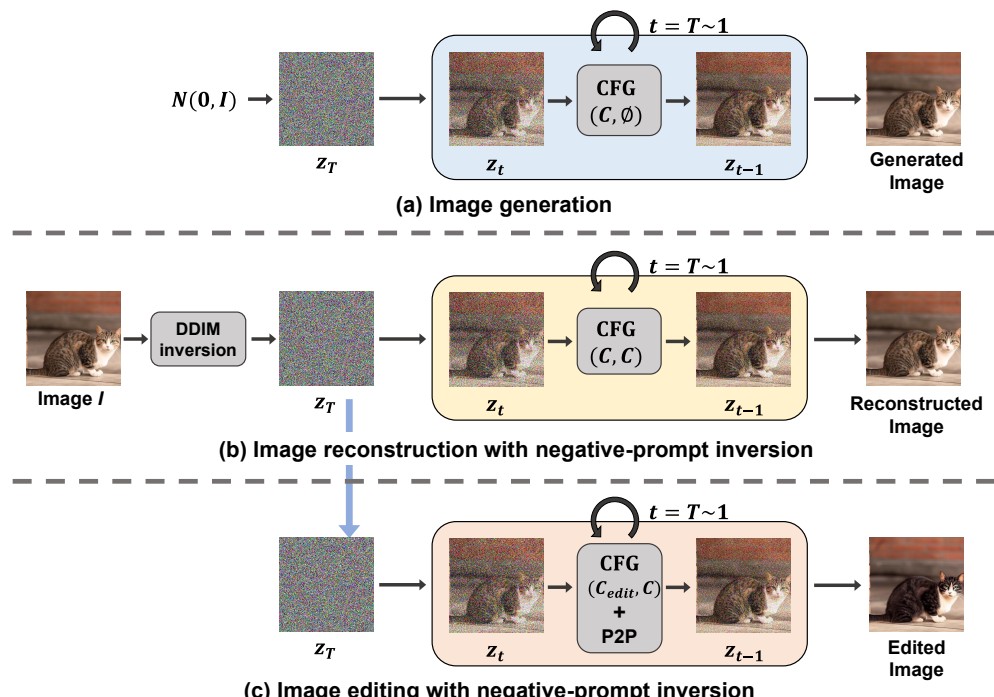

Figure 2: **Illustration of our framework.** (a) Image generation with CFG. A random noise $z_T$ is sampled from a standard normal distribution $\mathcal{N}(\mathbf{0}, \mathbf{I})$, then denoising $z_t$ with CFG over diffusion steps from $T$ to 1. $\mathrm{CFG}(C, \varnothing)$ denotes that using a prompt embedding $C$ for conditional prediction and the null-text embedding $\varnothing$ for unconditional prediction. (b) Image reconstruction with negative-prompt inversion. We replace the null-text embedding $\varnothing$ with the prompt embedding $C$ in CFG. (c) Image editing with negative-prompt inversion. We use the edited prompt embedding $C_{\mathrm{edit}}$ as the text condition and use the original prompt embedding $C$ instead of the null-text $\varnothing$ in CFG with an image editing method such as prompt-to-prompt (P2P).

is used as a reference for unconditional prediction to enhance the conditioning:

$$\tilde{\epsilon}_\theta(z_t, t, C, \varnothing) = \epsilon_\theta(z_t, t, \varnothing) + w\left(\epsilon_\theta(z_t, t, C) - \epsilon_\theta(z_t, t, \varnothing)\right), \tag{1}$$

where the guidance scale $w \geq 0$ controls strength of the condition.

In the inference phase, DDIM (Song et al., 2021a) calculates from the latent variable $z_t$ the latent variable $z_{t-1}$ via

$$z_{t-1} = \sqrt{\frac{\alpha_{t-1}}{\alpha_t}} z_t + \sqrt{\alpha_{t-1}} \left(\sqrt{\frac{1}{\alpha_{t-1}} - 1} - \sqrt{\frac{1}{\alpha_t} - 1}\right) \epsilon_\theta(z_t, t, C), \tag{2}$$

where $\boldsymbol{\alpha} := (\alpha_1, \ldots, \alpha_T) \in \mathbb{R}^T_{\geq 0}$ are hyper-parameters to determine noise scales at $T$ diffusion steps. The forward process can be represented in terms of $\epsilon_\theta(z_t, t, C)$ by inverting the reverse diffusion process (DDIM inversion) (Song et al., 2021a; Dhariwal & Nichol, 2021), as

$$z_{t+1} = \sqrt{\frac{\alpha_{t+1}}{\alpha_t}} z_t + \sqrt{\alpha_{t+1}} \left(\sqrt{\frac{1}{\alpha_{t+1}} - 1} - \sqrt{\frac{1}{\alpha_t} - 1}\right) \epsilon_\theta(z_t, t, C). \tag{3}$$

### 3.3 Null-text inversion

DDIM is known to work well: Given an original sample, by performing the forward process starting from the representation $z_0$ of the sample to obtain $z_T$ and then by inverting the

forward process, one can reconstruct the original sample with high quality when we do not use CFG (i.e., $w = 1$). However, since CFG is useful to strengthen the text conditioning, it is necessary to reconstruct original samples well even when one uses CFG (i.e., $w > 1$). Null-text inversion enables us to faithfully reconstruct given samples even when using CFG by optimizing the null-text embedding $\varnothing$ at each diffusion step $t$.

First, we calculate the sequence of latent variables $(\boldsymbol{z}_t^*)_{t \in \{1, \cdots, T\}}$ from $\boldsymbol{z}_0$ via DDIM inversion. Next, we do initialization with $\bar{\boldsymbol{z}}_T = \boldsymbol{z}_T^*$ and $\varnothing_T = \varnothing$. We then iteratively optimize $\varnothing_t$ for $t = T$ to 1 as follows: At each diffusion step $t$, assuming that we have $\bar{\boldsymbol{z}}_t$, one can calculate $\boldsymbol{z}_{t-1}$ via DDIM (13) and CFG (1) with the null-text embedding $\varnothing_t$ as

$$\boldsymbol{z}_{t-1}(\bar{\boldsymbol{z}}_t, t, C, \varnothing_t) = \sqrt{\frac{\alpha_{t-1}}{\alpha_t}} \bar{\boldsymbol{z}}_t + \sqrt{\alpha_{t-1}} \left( \sqrt{\frac{1}{\alpha_{t-1}} - 1} - \sqrt{\frac{1}{\alpha_t} - 1} \right) \tilde{\boldsymbol{\epsilon}}_\theta(\bar{\boldsymbol{z}}_t, t, C, \varnothing_t). \quad (4)$$

Then, we optimize $\varnothing_t$ to minimize the MSE between the predicted $\boldsymbol{z}_{t-1}(\bar{\boldsymbol{z}}_t, t, C, \varnothing_t)$ and $\boldsymbol{z}_{t-1}^*$:

$$\min_{\varnothing_t} \| \boldsymbol{z}_{t-1}(\bar{\boldsymbol{z}}_t, t, C, \varnothing_t) - \boldsymbol{z}_{t-1}^* \|_2^2,$$

with the initialization $\varnothing_t = \varnothing_{t+1}$. After several updates (e.g., 10 iterations), we fix $\varnothing_t$ and set $\bar{\boldsymbol{z}}_{t-1} = \boldsymbol{z}_{t-1}(\bar{\boldsymbol{z}}_t, t, C, \varnothing_t)$. By performing the optimization at $t = T, \ldots, 1$ sequentially, we can reconstruct the original image with high quality even when using CFG with $w > 1$.

### 3.4 NEGATIVE-PROMPT INVERSION

The proposed method, **negative-prompt inversion**, utilizes the text prompt embeddings $C$ instead of the optimized null-text embeddings $(\varnothing_t)_{t \in \{1, \ldots, T\}}$ in null-text inversion. As a result, we can perform reconstruction with only forward computation without optimization, significantly reducing computation time.

We now discuss how one can avoid optimization in our proposal, by more closely investigating the process of null-text inversion. Let us assume, for the following argument by induction, that at diffusion step $t$ in null-text inversion one has $\bar{\boldsymbol{z}}_t$ that is close enough to $\boldsymbol{z}_t^*$, so that one can regard $\bar{\boldsymbol{z}}_t = \boldsymbol{z}_t^*$ to hold. In null-text inversion, one obtains $\boldsymbol{z}_{t-1}$ from $\bar{\boldsymbol{z}}_t$ by moving one diffusion step backward using (4). Recall that $\boldsymbol{z}_t^*$ was calculated from $\boldsymbol{z}_{t-1}^*$ by moving one diffusion step forward in the diffusion process using (3):

$$\boldsymbol{z}_t^* = \sqrt{\frac{\alpha_t}{\alpha_{t-1}}} \boldsymbol{z}_{t-1}^* + \sqrt{\alpha_t} \left( \sqrt{\frac{1}{\alpha_t} - 1} - \sqrt{\frac{1}{\alpha_{t-1}} - 1} \right) \boldsymbol{\epsilon}_\theta(\boldsymbol{z}_{t-1}^*, t-1, C).$$

As we have assumed $\bar{\boldsymbol{z}}_t = \boldsymbol{z}_t^*$, one can substitute the above into (4), yielding

$$\bar{\boldsymbol{z}}_{t-1} = \boldsymbol{z}_{t-1}^* + \sqrt{\alpha_{t-1}} \left( \sqrt{\frac{1}{\alpha_{t-1}} - 1} - \sqrt{\frac{1}{\alpha_t} - 1} \right) \left( \tilde{\boldsymbol{\epsilon}}_\theta(\bar{\boldsymbol{z}}_t, t, C, \varnothing_t) - \boldsymbol{\epsilon}_\theta(\boldsymbol{z}_{t-1}^*, t-1, C) \right). \quad (5)$$

It implies that the discrepancy between $\bar{\boldsymbol{z}}_{t-1}$ and $\boldsymbol{z}_{t-1}^*$ in null-text inversion will be minimized when the predicted noises are equal:

$$\boldsymbol{\epsilon}_\theta(\boldsymbol{z}_{t-1}^*, t-1, C) = \tilde{\boldsymbol{\epsilon}}_\theta(\bar{\boldsymbol{z}}_t, t, C, \varnothing_t)$$
$$= w \boldsymbol{\epsilon}_\theta(\bar{\boldsymbol{z}}_t, t, C) + (1-w) \boldsymbol{\epsilon}_\theta(\bar{\boldsymbol{z}}_t, t, \varnothing_t)$$

If furthermore we are allowed to assume that the predicted noises at adjacent diffusion steps are equal, i.e., $\boldsymbol{\epsilon}_\theta(\boldsymbol{z}_{t-1}^*, t-1, C) = \boldsymbol{\epsilon}_\theta(\boldsymbol{z}_t^*, t, C) = \boldsymbol{\epsilon}_\theta(\bar{\boldsymbol{z}}_t, t, C)$, then we can deduce that at the optimum the conditional and unconditional predictions are equal:

$$\boldsymbol{\epsilon}_\theta(\bar{\boldsymbol{z}}_t, t, C) = \boldsymbol{\epsilon}_\theta(\bar{\boldsymbol{z}}_t, t, \varnothing_t) \quad (6)$$

Therefore, the optimized $\varnothing_t$ can be approximated by the prompt embedding $C$, so that we can discard the optimization of the null-text embedding $\varnothing_t$ in null-text inversion altogether, simply by replacing the null-text embedding $\varnothing_t$ with $C$. See Appendix A for more detail on a theoretical justification and empirical validation in practical settings. The argument so far has the following two consequences:

1. For reconstruction, letting $\varnothing_t = C$ amounts to not using CFG at all (since $\hat{\epsilon}_\theta(\boldsymbol{z}_t, t, C, C) = \epsilon_\theta(\boldsymbol{z}_t, t, C)$ holds for any $w$). The above argument can thus be regarded as providing a justification to the empirically well-known observation that DDIM works well without CFG.

2. For editing, optimizing $\varnothing_t$ in null-text inversion can be replaced by the simple substitution $\varnothing_t = C_{\mathrm{src}}$ and $C = C_{\mathrm{edit}}$ during the sampling process, where $C_{\mathrm{src}}$ and $C_{\mathrm{edit}}$ denote an embedding of a source prompt and an edited prompt respectively.

Figure 2 illustrates our framework. (a) represents the image generation using CFG, while (b) represents our proposal, negative-prompt inversion, which replaces the null-text embedding with the input prompt embedding $C$. Additionally, in case of image editing like prompt-to-prompt (P2P), we can set the embedding $C_{\mathrm{edit}}$ of edited prompt as the text condition and set the original prompt embedding $C$ as the negative-prompt embedding instead of the null-text embedding, as shown in Fig. 2 (c).

## 4 EXPERIMENTS

### 4.1 SETTING

In this section, we evaluate the proposed method qualitatively and quantitatively. We experimented it using Stable Diffusion v1.5 in 🤗 Diffusers (von Platen et al., 2022) implemented with PyTorch (Paszke et al., 2019). Our code used in the experiments is provided in SM. Following Mokady et al. (2023), we used 100 images and captions, randomly selected from validation data in COCO dataset (Lin et al., 2014), in our experiments. The images were trimmed to make them square and resized to $512 \times 512$. Unless otherwise specified, in both DDIM inversion and sampling we set the number of the sampling steps to be 50 via using the stride of 20 over the $T = 1000$ diffusion steps.

We compared our method with DDIM inversion followed by DDIM sampling with CFG and null-text inversion, and evaluated its reconstruction quality by peak signal-to-noise ratio (PSNR) and learned perceptual image patch similarity (LPIPS) (Zhang et al., 2018), whereas its editing quality by CLIP score (Radford et al., 2021). See Appendix B for our setting of null-text inversion. The inference speed was measured on one NVIDIA RTX A6000 connected to one AMD EPYC 7343 (16 cores, 3.2 GHz clockspeed).

### 4.2 RECONSTRUCTION

The left three columns of Table 1 shows PSNR, LPIPS, and inference time of reconstruction by the three methods compared. In terms of PSNR (higher is better) and LPIPS (lower is better), the reconstruction quality of the proposed method was slightly worse than that of null-text inversion but far better than that of DDIM inversion. On the other hand, the inference speed was 30 times as fast as null-text inversion. This remarkable acceleration is achieved since the iterative optimization and backpropagation processing required for null-text inversion are not necessary for our method.

In Figure 3, the left four columns display examples of reconstruction by the three methods. DDIM inversion reconstructed images with noticeable differences from the input images, such as object position and shape. In contrast, null-text inversion and negative-prompt inversion (Ours) were capable of reconstructing images that are nearly identical to the input images, and the proposed method achieved a high reconstruction quality comparable to that of null-text inversion. See Appendix C.1 for additional reconstruction examples. These results suggest that the proposed method can achieve reconstruction quality nearly equivalent to null-text inversion, with a speedup of over 30 times. Additionally, we also measured the memory usage of the three methods, and found that our method and DDIM inversion used half as much memory as null-text inversion.

Table 1: **Evaluation of reconstruction/editing quality and speed in each method.** $\pm$ represents 95% confidence intervals. Note that as DDIM inversion and ours perform the same process, they are theoretically at the same speed.

| Method | PSNR↑ | LPIPS↓ | Speed (s) | CLIP↑ |
|---|---|---|---|---|
| Imagic | $17.17 \pm 0.66$ | $0.356 \pm 0.025$ | $552.86 \pm 0.16$ | $22.99 \pm 0.77$ |
| DDIM inversion | $14.05 \pm 0.34$ | $0.528 \pm 0.022$ | $\mathbf{4.61 \pm 0.03}$ | $\mathbf{25.10 \pm 0.74}$ |
| Null-text inversion | $\mathbf{26.11 \pm 0.81}$ | $\mathbf{0.075 \pm 0.007}$ | $129.77 \pm 2.97$ | $24.07 \pm 0.72$ |
| **Ours** | $23.38 \pm 0.66$ | $0.160 \pm 0.016$ | $\mathbf{4.63 \pm 0.02}$ | $23.77 \pm 0.74$ |

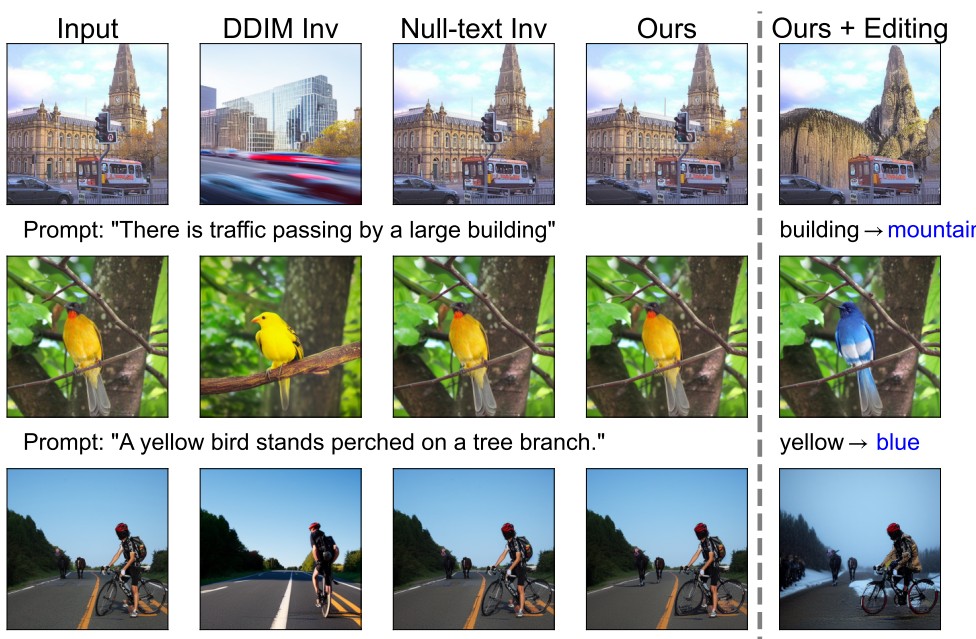

Figure 3: **Evaluation of reconstructed images.** The left 4 columns show the reconstruction results of each method, and the right column shows the image editing results using our method and prompt-to-prompt. The editing prompts are described below the edited images, that were created by replacing words or adding new words to the original prompt. Our method reconstructs input images as well as null-text inversion and edited images also preserve the structure of the input images.

## 4.3 Editing

We next demonstrate the feasibility of editing real images by combining our inversion method with existing image editing methods. Our method is independent of the image editing approach and is principally compatible with any method that uses CFG, allowing for the selection of an appropriate image editing method depending on the objective. Here, we verify the effectiveness of our method for real-image editing using prompt-to-prompt (Hertz et al., 2023) in the same manner as in Mokady et al. (2023).

The rightmost column of Table 1 shows CLIP scores of editing results by prompt-to-prompt with the three methods compared. Taking account of the standard errors, one can see that the proposed method and null-text inversion achieved almost the same CLIP scores. Although the score of DDIM inversion was the best, by considering the scores in conjunction with reconstruction quality, the editing quality of the proposed method was comparable to null-text inversion. In addition, we also compared our method with Imagic (Kawar et al., 2023) as another editing method, the editing quality of the proposed method was also better than that of Imagic. For qualitative evaluation, the rightmost column of Fig. 3 shows

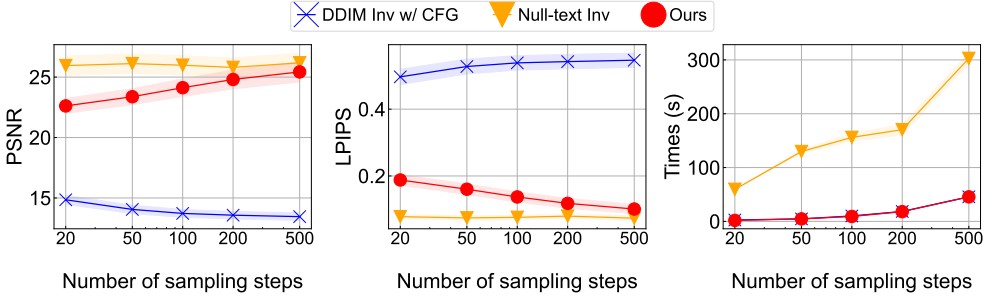

Figure 4: **Reconstruction quality and speed versus the number of sampling steps.** Higher PSNR is better (left), lower LPIPS is better (middle), and shorter execution time is better (right). Shadings indicate 95% confidence intervals.

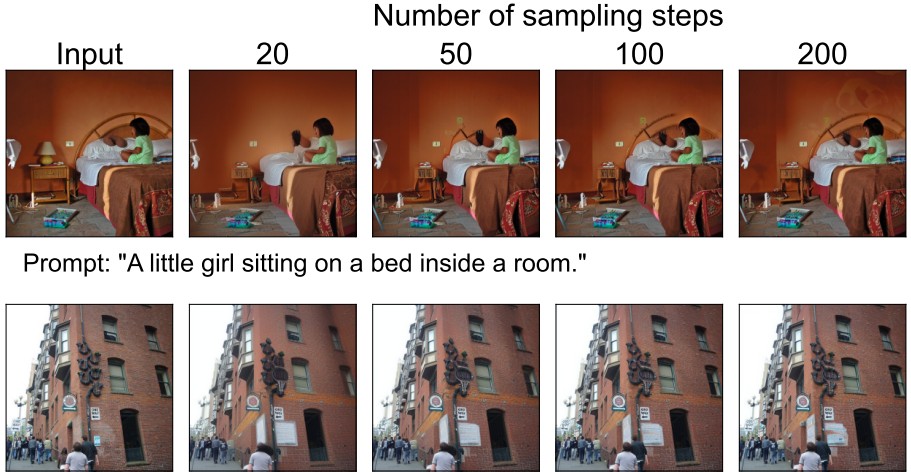

Figure 5: **Reconstructed images when changing the number of sampling steps.** The images become more similar to the input images as the number of sampling steps increases.

examples of real-image editing via prompt-to-prompt using the proposed method. The proposed method managed to maintain the composition while editing the image according to the modified prompt, such as replacing the objects and changing the background. Additional editing examples are provided in Appendices C.2 and C.3. These observations show that our inversion method can be combined with editing methods like prompt-to-prompt to enable rapid real-image editing.

### 4.4 NUMBER OF SAMPLING STEPS

As the proposed method allows fast reconstruction/editing, one may be able to use a larger number of sampling steps to further improve reconstruction quality, at the expense of reduced speed. To investigate the relationship between the number of sampling steps and reconstruction quality, we measured the PSNR and LPIPS using five different sampling steps: 20, 50, 100, 200, and 500.

Figure 4 shows PSNR, LPIPS, and speed versus the number of sampling steps by the three methods. Although results with high enough quality were obtained with 50 sampling steps, increasing the number of sampling steps further improved the reconstruction quality of the proposed method, approaching that by null-text inversion. It should be noted that the total execution time is roughly given by the product of the execution time per sampling step and

the number of sampling steps so that even if the proposed inversion method is performed with 500 sampling steps, it would still take less time than executing null-text inversion with 50 sampling steps thanks to the $30\times$ speedup. In fact, Figure 4 right shows the time taken for inversion; with 500 sampling steps, it took 46 seconds, which is approximately three times faster than the null-text inversion with 50 sampling steps, which took 130 seconds. We would like to note that in Fig. 4 right the execution time of null-text inversion was not proportional to the number of sampling steps, since in our experimental setting the early stopping employed in the null-text optimization was more effective as the number of sampling steps became larger.

Figure 5 describes how the reconstructed image changed as the number of sampling steps was increased. Even with a small number of sampling steps, such as 20, the input image's objects and composition were successfully reconstructed. Focusing on the finer details, for example, the head of the bed and the desk in the first row, and the wall color and pipes on the wall in the second row, we observe that the reconstruction quality improved as the number of sampling steps was increased. This improvement is generally imperceptible at first glance, suggesting that conventionally adopted numbers of sampling steps, such as 20 and 50 sampling steps, yield sufficiently satisfactory reconstruction results for practical purposes.

## 5 Limitations

A limitation of the proposed method is that the average reconstruction quality does not reach that of null-text inversion. As demonstrated in the previous section, the difference is generally imperceptible at first glance; however, there were instances where our inversion method failed significantly. For example, we observed that the proposed method tended to fail in reconstructing persons. Such failures could be attributed to characteristics of Stable Diffusion's AutoEncoder, which struggles to reconstruct human faces. In such cases, employing a more effective encoder-decoder pair may result in improvements. Moreover, some failure cases were improved by increasing the number of sampling steps. Failure images are presented in Appendix C.4.

Although failures in post-reconstruction image editing may occur, our inversion method is independent of editing methods, making the related discussion beyond the scope of this paper.

## 6 Conclusions

We have proposed negative-prompt inversion, which enables real-image inversion in diffusion models without the need for optimization. Experimentally, it produced visually high-quality reconstruction results comparable to inversion methods requiring optimization, while achieving a remarkable speed-up of over 30 times. Furthermore, we discovered that increasing the number of sampling steps further improved the reconstruction quality while maintaining faster computational time than existing methods.

On the basis of these results, our method provides a practical approach for real-image reconstruction. This utility excels in high-computational-cost scenarios, such as video editing, where our method proves to be even more beneficial. Moreover, by parallelizing multiple GPUs and optimizing the program, there is potential for our method to achieve higher throughput and lower latency, where even the real-time processing would be possible. Although the proposed approach reduces computational costs and is available to any user, it does not encourage socially inappropriate use.

## 7 Reproducibility Statement

We provide the details of our method and experimental settings in the appendices to help with reproducibility. In Appendix A, we provide the theoretical details of the derivation of our method with empirical validation of the derivation. Appendix B provides the imple-

mentation details of null-text inversion used in our experiments. Additionally, our code is also made available in Supplemental Materials.

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

# A Justifying arguments

## A.1 Theoretical consideration

In this appendix, we firstly provide a continuous-time description of DDPM and DDIM processes. We start with the stochastic differential equation describing continuous-time random diffusion of particles in a $D$-dimensional space:

$$d\boldsymbol{z} = -\gamma_t \boldsymbol{z} + \sqrt{2\gamma_t} d\boldsymbol{W}, \tag{7}$$

where $W$ is the $D$-dimensional Wiener process and where $\gamma_t > 0$ is a deterministic and integrable function of $t$. If one lets $\gamma_t$ to be independent of $t$, then (7) describes what is called the Ornstein-Urlenbech (OU) process, so that (7) can be regarded as a generalized version of the OU process. The distribution $p_t(\boldsymbol{z})$ of the random particles following the diffusion process (7) at time $t$ is known to follow the Fokker-Planck equation

$$\frac{\partial p_t}{\partial t} = \gamma_t \{\nabla(\boldsymbol{z} p_t) + \Delta p_t\} \tag{8}$$

The solution of (8) given the initial condition $p_0(\boldsymbol{z}) = \delta(\boldsymbol{z}|_{t=0} - \boldsymbol{z}_0)$, i.e., all the random particles are located at the position $\boldsymbol{z}_0$ at time 0, or equivalently, one starts the diffusion process with a sample located at $\boldsymbol{z}_0$, is evaluated as

$$p_t(\boldsymbol{z} \mid \boldsymbol{z}|_{t=0} = \boldsymbol{z}_0) = \mathcal{N}\left(\sqrt{\alpha_t}\boldsymbol{z}_0, (1 - \alpha_t)\boldsymbol{I}\right), \tag{9}$$

where

$$\alpha_t := \exp\left(-\int_0^t \gamma_s \, ds\right). \tag{10}$$

We also write it as

$$\boldsymbol{z}_t|_{\boldsymbol{z}_0} \sim \mathcal{N}\left(\sqrt{\alpha_t}\boldsymbol{z}_0, (1 - \alpha_t)\boldsymbol{I}\right). \tag{11}$$

Furthermore, for $s \leq t$, the conditional distribution of the particles at time $t$ conditional on the particle located at $\boldsymbol{z}_s$ at time $s$ is given by

$$\boldsymbol{z}_t|_{\boldsymbol{z}_s} \sim \mathcal{N}\left(\sqrt{\frac{\alpha_t}{\alpha_s}}\boldsymbol{z}_s, \left(1 - \frac{\alpha_t}{\alpha_s}\right)\boldsymbol{I}\right). \tag{12}$$

Comparing these formulas with those in Ho et al. (2020, Section 2) reveals that discretizing the above process in time will give us the formulation of DDPM.

Assuming that the Fokker-Planck equation (8) is given, the corresponding random process is not unique, and there are several other random processes which are consistent with (8) than the above generalized OU process (7). For example, we may take a specific time instant $t = T > 0$ and require the particle position $\boldsymbol{z}_T$ at time $T$ given the initial position $\boldsymbol{z}_0$ at time $t = 0$ to follow the Gaussian distribution

$$\boldsymbol{z}_T|_{\boldsymbol{z}_0} \sim \mathcal{N}\left(\sqrt{\alpha_T}\boldsymbol{z}_0, (1 - \alpha_T)\boldsymbol{I}\right), \tag{13}$$

and then determine the particle position $\boldsymbol{z}_t$ at any time $t \geq 0$ as

$$\boldsymbol{z}_t = \sqrt{\frac{1 - \alpha_t}{1 - \alpha_T}}\boldsymbol{z}_T + \left(\sqrt{\alpha_t} - \sqrt{\frac{\alpha_T}{1 - \alpha_T}}\sqrt{1 - \alpha_t}\right)\boldsymbol{z}_0. \tag{14}$$

One can then confirm that the conditional distribution of the particle position $\boldsymbol{z}_t$ at time $t$ conditional on $\boldsymbol{z}_0$ is given by $\mathcal{N}\left(\sqrt{\alpha_t}\boldsymbol{z}_0, (1 - \alpha_t)\boldsymbol{I}\right)$, which demonstrates that the distribution of the particles following the above process also satisfies the same Fokker-Planck equation (8). One can furthermore show that discretizing the above process in time will give us the formulation of DDIM (Song et al., 2021a).

When considering $\boldsymbol{z}_t$ given $\boldsymbol{z}_0$ and $\boldsymbol{z}_T$, let $\boldsymbol{d}_t := (\boldsymbol{z}_t - \sqrt{\alpha_t}\boldsymbol{z}_0)/\sqrt{1 - \alpha_t}$ be the normalized noise component in $\boldsymbol{z}_t$ relative to $\sqrt{\alpha_t}\boldsymbol{z}_0$. One can show, by rearranging terms in (14), that $\boldsymbol{d}_t = \boldsymbol{d}_T$ holds for any $t$. Letting $\boldsymbol{d} := \boldsymbol{d}_t$ due to the independence of $\boldsymbol{d}_t$ on $t$, one can furthermore show, via (13), that $\boldsymbol{d}$ given $\boldsymbol{z}_0$ follows the standard Gaussian distribution $\mathcal{N}(\boldsymbol{0}, \boldsymbol{I})$. In other words, given $\boldsymbol{z}_0$ and $\boldsymbol{z}_T$, the normalized noise component $\boldsymbol{d}_t$ in DDIM does not depend on $t$. Therefore, the diffusion paths in DDIM are straight half-lines $\{\boldsymbol{z}_t =$

$\sqrt{\alpha_t}\boldsymbol{z}_0 + \sqrt{1-\alpha_t}\boldsymbol{d} : t \geq 0, \boldsymbol{d} \sim \mathcal{N}(\boldsymbol{0}, \boldsymbol{I})\}$ starting from $\boldsymbol{z}_0$ with random velocity $\boldsymbol{d} \sim \mathcal{N}(\boldsymbol{0}, \boldsymbol{I})$.

Assuming that $\boldsymbol{z}_t$ is available, the model $\boldsymbol{\epsilon}_\theta(\boldsymbol{z}_t, t)$ attempts to estimate the velocity $\boldsymbol{d}_t$ from $\boldsymbol{z}_t$, which in turn yields an estimate $f_\theta^{(t)}(\boldsymbol{z}_t) := (\boldsymbol{z}_t - \sqrt{1-\alpha_t}\boldsymbol{\epsilon}_\theta(\boldsymbol{z}_t, t))/\sqrt{\alpha_t}$ of $\boldsymbol{z}_0$, and then one can use it to estimate $\boldsymbol{z}_s$ for any $s$ by plugging it into the equality $\boldsymbol{d}_t = \boldsymbol{d}_s$. Specifically, $\boldsymbol{z}_s$ is estimated via

$$\begin{aligned}
\boldsymbol{z}_s &= \sqrt{\alpha_s}\boldsymbol{z}_0 + \sqrt{1-\alpha_s}\frac{\boldsymbol{z}_t - \sqrt{\alpha_t}\boldsymbol{z}_0}{\sqrt{1-\alpha_t}} \\
&\approx \sqrt{\alpha_s}\left(\frac{\boldsymbol{z}_t - \sqrt{1-\alpha_t}\boldsymbol{\epsilon}_\theta(\boldsymbol{z}_t, t)}{\sqrt{\alpha_t}}\right) + \sqrt{1-\alpha_s}\boldsymbol{\epsilon}_\theta(\boldsymbol{z}_t, t) \\
&= \sqrt{\frac{\alpha_s}{\alpha_t}}\boldsymbol{z}_t + \sqrt{\alpha_s}\left(\sqrt{\frac{1-\alpha_s}{\alpha_s}} - \sqrt{\frac{1-\alpha_t}{\alpha_t}}\right)\boldsymbol{\epsilon}_\theta(\boldsymbol{z}_t, t).
\end{aligned} \tag{15}$$

When one takes $s = t \pm 1$, the above formula is reduced to

$$\boldsymbol{z}_{t\pm 1} \approx \sqrt{\frac{\alpha_{t\pm 1}}{\alpha_t}}\boldsymbol{z}_t + \sqrt{\alpha_{t\pm 1}}\left(\sqrt{\frac{1}{\alpha_{t\pm 1}} - 1} - \sqrt{\frac{1}{\alpha_t} - 1}\right)\boldsymbol{\epsilon}_\theta(\boldsymbol{z}_t, t), \tag{16}$$

which corresponds to (3) and (13) in the main text.

The argument presented so far is based on conditioning on sample $\boldsymbol{z}_0$, which is not justifiable in the actual process of DDIM sampling where there exists more than one sample and where the model does not look at $\boldsymbol{z}_0$. We thus extend the above argument via assuming $\boldsymbol{z}_0$ to be generated according to a certain probability distribution $p(\boldsymbol{z}_0)$. More concretely, we assume $\boldsymbol{z}_0 \sim p(\boldsymbol{z}_0)$ and $\boldsymbol{d} \sim \mathcal{N}(\boldsymbol{0}, \boldsymbol{I})$, which induces the diffusion path $\boldsymbol{z}_t = \sqrt{\alpha_t}\boldsymbol{z}_0 + \sqrt{1-\alpha_t}\boldsymbol{d}$, $t \geq 0$, in DDIM according to the above discussion. Consequently, at position $\boldsymbol{z}$ and at time $t$, the "velocity field" $\boldsymbol{\epsilon}(\boldsymbol{z}, t)$ to be learned by the model $\boldsymbol{\epsilon}_\theta(\boldsymbol{z}, t)$ is not determined by a single sample $\boldsymbol{z}_0$ but given by the posterior mean of $\boldsymbol{d} = (\boldsymbol{z} - \sqrt{\alpha_t}\boldsymbol{z}_0)/\sqrt{1-\alpha_t}$ with respect to the posterior distribution of $\boldsymbol{z}_0$ given $\boldsymbol{z}$, which is obtained from the prior distributions $\boldsymbol{z}_0 \sim p(\boldsymbol{z}_0)$ and $\boldsymbol{d} \sim \mathcal{N}(\boldsymbol{0}, \boldsymbol{I})$, as well as the likelihood $p(\boldsymbol{z} \mid \boldsymbol{z}_0, \boldsymbol{d}) = \delta(\boldsymbol{z} - \sqrt{\alpha_t}\boldsymbol{z}_0 - \sqrt{1-\alpha_t}\boldsymbol{d})$.

**Proposition 1.** *Assume $\boldsymbol{z}_0 \sim p(\boldsymbol{z}_0)$ and $\boldsymbol{d} \sim \mathcal{N}(\boldsymbol{0}, \boldsymbol{I})$. Then the velocity field $\boldsymbol{\epsilon}(\boldsymbol{z}, t)$ in DDIM at position $\boldsymbol{z}$ and at time $t$, which is to be learned by the model $\boldsymbol{\epsilon}_\theta(\boldsymbol{z}, t)$, is given by*

$$\boldsymbol{\epsilon}(\boldsymbol{z}, t) = \frac{\left\langle \frac{\boldsymbol{z} - \sqrt{\alpha_t}\boldsymbol{z}_0}{\sqrt{1-\alpha_t}}p_G\left(\frac{\boldsymbol{z} - \sqrt{\alpha_t}\boldsymbol{z}_0}{\sqrt{1-\alpha_t}}\right)\right\rangle_{\boldsymbol{z}_0}}{\left\langle p_G\left(\frac{\boldsymbol{z} - \sqrt{\alpha_t}\boldsymbol{z}_0}{\sqrt{1-\alpha_t}}\right)\right\rangle_{\boldsymbol{z}_0}}, \tag{17}$$

*where*

$$p_G(\boldsymbol{d}) = \frac{1}{(2\pi)^{D/2}}e^{-\|\boldsymbol{d}\|_2^2/2} \tag{18}$$

*denotes the probability density function of the D-dimensional standard Gaussian distribution, and where $\langle \cdot \rangle_{\boldsymbol{z}_0}$ denotes expectation with respect to $\boldsymbol{z}_0 \sim p(\boldsymbol{z}_0)$.*

*Proof.* The joint distribution of $\boldsymbol{z}_0$ and $\boldsymbol{z}$ is given by

$$\begin{aligned}
p(\boldsymbol{z}_0, \boldsymbol{z}) &= \int p(\boldsymbol{z} \mid \boldsymbol{z}_0, \boldsymbol{d})p(\boldsymbol{z}_0)p_G(\boldsymbol{d})\, d\boldsymbol{d} \\
&= \int \delta(\boldsymbol{z} - \sqrt{\alpha_t}\boldsymbol{z}_0 - \sqrt{1-\alpha_t}\boldsymbol{d})p(\boldsymbol{z}_0)p_G(\boldsymbol{d})\, d\boldsymbol{d} \\
&= p_G\left(\frac{\boldsymbol{z} - \sqrt{\alpha_t}\boldsymbol{z}_0}{\sqrt{1-\alpha_t}}\right)p(\boldsymbol{z}_0),
\end{aligned} \tag{19}$$

from which the posterior distribution of $\boldsymbol{z}_0$ given $\boldsymbol{z}$ is obtained as

$$p(\boldsymbol{z}_0 \mid \boldsymbol{z}) = \frac{p_G\left(\frac{\boldsymbol{z} - \sqrt{\alpha_t}\boldsymbol{z}_0}{\sqrt{1-\alpha_t}}\right)p(\boldsymbol{z}_0)}{\left\langle p_G\left(\frac{\boldsymbol{z} - \sqrt{\alpha_t}\boldsymbol{z}_0}{\sqrt{1-\alpha_t}}\right)\right\rangle_{\boldsymbol{z}_0}} \tag{20}$$

The velocity $\boldsymbol{\epsilon}(\boldsymbol{z}, t)$ at $\boldsymbol{z}$ and $t$, to be learned by the model, is given by the posterior mean of $\boldsymbol{d} = (\boldsymbol{z} - \sqrt{\alpha_t}\boldsymbol{z}_0)/\sqrt{1-\alpha_t}$, which is represented as (17), proving the proposition. $\square$

Despite its complex appearance, one can see that the velocity field $\boldsymbol{\epsilon}(\boldsymbol{z}, t)$ in (17) is continuous in $\boldsymbol{z}$ and $t > 0$. This continuity implies that, for $t, s > 0$, when $|\alpha_t - \alpha_s|$ and $\|\boldsymbol{z} - \boldsymbol{z}'\|$ are small, one can expect $\boldsymbol{\epsilon}(\boldsymbol{z}, t) \approx \boldsymbol{\epsilon}(\boldsymbol{z}', s)$ to hold.

In what follows, we provide a justifying argument for the proposed method, via extending the argument so far by incorporating conditioning into the model. It is straightforward to incorporate conditioning in the DDIM inversion formula (3) and the DDIM sampling formula (13), by replacing the model $\boldsymbol{\epsilon}_\theta(\boldsymbol{z}_t, t)$ without conditioning with the conditional model $\boldsymbol{\epsilon}_\theta(\boldsymbol{z}_t, t, C)$, where $C$ is the prompt embedding. In various applications, on the other hand, the reverse process using the DDIM sampling formula (3) is often combined with CFG to strengthen the effects of the conditioning, where the conditional model $\boldsymbol{\epsilon}_\theta(\boldsymbol{z}_t, t, C)$ is further replaced with

$$\tilde{\boldsymbol{\epsilon}}_\theta(\boldsymbol{z}_t, t, C, \varnothing) = \boldsymbol{\epsilon}_\theta(\boldsymbol{z}_t, t, \varnothing) + w\left(\boldsymbol{\epsilon}_\theta(\boldsymbol{z}_t, t, C) - \boldsymbol{\epsilon}_\theta(\boldsymbol{z}_t, t, \varnothing)\right), \tag{21}$$

where $w \geq 0$ is the guidance scale, which controls the strength of the conditioning, and where $\varnothing$ is the null-text embedding.

The first step of null-text inversion is to obtain $\boldsymbol{z}_t^*$ for $t = 1, \ldots, T$ by initializing $\boldsymbol{z}_0^* = \boldsymbol{z}_0$ and successively applying the forward process derived as the DDIM inversion formula:

$$\boldsymbol{z}_t^* = \sqrt{\frac{\alpha_t}{\alpha_{t-1}}}\boldsymbol{z}_{t-1}^* + \sqrt{\alpha_t}\left(\sqrt{\frac{1}{\alpha_t} - 1} - \sqrt{\frac{1}{\alpha_{t-1}} - 1}\right)\boldsymbol{\epsilon}_\theta(\boldsymbol{z}_{t-1}^*, t-1, C), \tag{22}$$

where the model $\boldsymbol{\epsilon}_\theta(\boldsymbol{z}_{t-1}, t-1)$ in (3) without conditioning has been replaced with the conditional model $\boldsymbol{\epsilon}_\theta(\boldsymbol{z}_{t-1}^*, t-1, C)$.

Next, starting from $\bar{\boldsymbol{z}}_T = \boldsymbol{z}_T^*$, we calculate the reverse diffusion process to obtain $\bar{\boldsymbol{z}}_t$ in the backward direction, while optimizing the null-text embedding $\varnothing_t$ at each diffusion step so that $\bar{\boldsymbol{z}}_t$ well reproduces $\boldsymbol{z}_t^*$. More specifically, for $t = T, T-1, \ldots, 1$, $\bar{\boldsymbol{z}}_{t-1}$ is calculated via combining the DDIM sampling (16) and CFG (21) as

$$\bar{\boldsymbol{z}}_{t-1} = \sqrt{\frac{\alpha_{t-1}}{\alpha_t}}\bar{\boldsymbol{z}}_t + \sqrt{\alpha_{t-1}}\left(\sqrt{\frac{1}{\alpha_{t-1}} - 1} - \sqrt{\frac{1}{\alpha_t} - 1}\right)\tilde{\boldsymbol{\epsilon}}_\theta(\bar{\boldsymbol{z}}_t, t, C, \varnothing_t). \tag{23}$$

The null-text embedding $\varnothing_t$ is optimized to minimize the MSE between $\boldsymbol{z}_{t-1}^*$ and $\bar{\boldsymbol{z}}_{t-1}$ as

$$\min_{\varnothing_t} \|\boldsymbol{z}_{t-1}^* - \bar{\boldsymbol{z}}_{t-1}\|_2^2, \tag{24}$$

where $\bar{\boldsymbol{z}}_{t-1}$ is dependent on $\varnothing_t$ via (23). The following proposition shows that the choice $\varnothing_t = C$ does minimize the MSE between $\boldsymbol{z}_{t-1}^*$ and $\bar{\boldsymbol{z}}_{t-1}$ under an ideal situation.

**Proposition 2.** *Assume that there is only one sample, and that the guidance scale $w$ in CFG is not equal to 1. For any $t$, if the model $\boldsymbol{\epsilon}(\boldsymbol{z}, t, C)$ is able to correctly predict the velocity field and if $\boldsymbol{z}_t^* = \bar{\boldsymbol{z}}_t$ holds true, then the difference between $\boldsymbol{z}_{t-1}^*$ and $\bar{\boldsymbol{z}}_{t-1}$ in null-text inversion is made equal to zero if and only if $\boldsymbol{\epsilon}_\theta(\bar{\boldsymbol{z}}_t, t, \varnothing_t)$ is equal to $\boldsymbol{\epsilon}_\theta(\bar{\boldsymbol{z}}_t, t, C)$.*

*Proof.* The difference between $\boldsymbol{z}_{t-1}^*$ and $\bar{\boldsymbol{z}}_{t-1}$ is expressed as

$$\boldsymbol{z}_{t-1}^* - \bar{\boldsymbol{z}}_{t-1} = \boldsymbol{z}_{t-1}^* - \sqrt{\frac{\alpha_{t-1}}{\alpha_t}}\bar{\boldsymbol{z}}_t - \sqrt{\alpha_{t-1}}\left(\sqrt{\frac{1}{\alpha_{t-1}} - 1} - \sqrt{\frac{1}{\alpha_t} - 1}\right)\tilde{\boldsymbol{\epsilon}}_\theta(\bar{\boldsymbol{z}}_t, t, C, \varnothing_t)$$

$$= \boldsymbol{z}_{t-1}^* - \sqrt{\frac{\alpha_{t-1}}{\alpha_t}}\boldsymbol{z}_t^* - \sqrt{\alpha_{t-1}}\left(\sqrt{\frac{1}{\alpha_{t-1}} - 1} - \sqrt{\frac{1}{\alpha_t} - 1}\right)\tilde{\boldsymbol{\epsilon}}_\theta(\bar{\boldsymbol{z}}_t, t, C, \varnothing_t)$$

$$= \sqrt{\alpha_{t-1}}\left(\sqrt{\frac{1}{\alpha_{t-1}} - 1} - \sqrt{\frac{1}{\alpha_t} - 1}\right)\left(\boldsymbol{\epsilon}_\theta(\boldsymbol{z}_{t-1}^*, t-1, C) - \tilde{\boldsymbol{\epsilon}}_\theta(\bar{\boldsymbol{z}}_t, t, C, \varnothing_t)\right). \tag{25}$$

In the second line of the above equation we used the assumption $\boldsymbol{z}_t^* = \bar{\boldsymbol{z}}_t$, and in the third line we substituted (22) into $\boldsymbol{z}_t^*$ above.

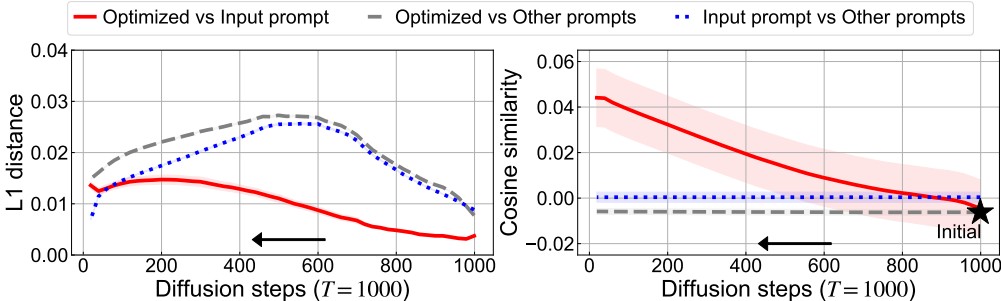

Figure 6: **Similarity between the optimized null-text and the input prompt.** (Left) The mean $L_1$ distance between predicted noises using the optimized null-text embedding and the input prompt. (Right) The mean similarity between the optimized null-text embedding and the input prompt. The solid line shows the similarity between the optimized null-text and the input prompt. The dashed line shows the similarity between the optimized null-text and the other prompts. The dotted line shows the similarity between the input prompt and the other prompts. **Initial** represents the starting point of optimization, and optimization is performed in the order indicated by the direction of the arrow in 50 sampling steps. Shaded regions indicate 95% confidence intervals.

As described above, the model $\boldsymbol{\epsilon}_\theta(\boldsymbol{z}_t, t, C)$ attempts to estimate noise $\boldsymbol{d}_t$ from $\boldsymbol{z}_t$, and the assumption that the model correctly predicts the noise, together with the discussion at the beginning of this section, implies that $\boldsymbol{\epsilon}_\theta(\boldsymbol{z}_t^*, t, C) = \boldsymbol{\epsilon}_\theta(\boldsymbol{z}_{t-1}^*, t-1, C)$ should hold. One therefore has

$$\begin{aligned}
\boldsymbol{\epsilon}_\theta(\boldsymbol{z}_{t-1}^*, t-1, C) - \tilde{\boldsymbol{\epsilon}}_\theta(\bar{\boldsymbol{z}}_t, t, C, \varnothing_t) &= \boldsymbol{\epsilon}_\theta(\boldsymbol{z}_t^*, t, C) - \tilde{\boldsymbol{\epsilon}}_\theta(\bar{\boldsymbol{z}}_t, t, C, \varnothing_t) \\
&= \boldsymbol{\epsilon}_\theta(\bar{\boldsymbol{z}}_t, t, C) - \tilde{\boldsymbol{\epsilon}}_\theta(\bar{\boldsymbol{z}}_t, t, C, \varnothing_t) \\
&= (1-w)\left(\boldsymbol{\epsilon}_\theta(\bar{\boldsymbol{z}}_t, t, C) - \boldsymbol{\epsilon}_\theta(\bar{\boldsymbol{z}}_t, t, \varnothing_t)\right). \qquad (26)
\end{aligned}$$

As we have assumed $w \neq 1$, $\boldsymbol{z}_{t-1}^* - \bar{\boldsymbol{z}}_{t-1}$ is proportional to $\boldsymbol{\epsilon}_\theta(\bar{\boldsymbol{z}}_t, t, C) - \boldsymbol{\epsilon}_\theta(\bar{\boldsymbol{z}}_t, t, \varnothing_t)$, and it is made equal to zero if and only if $\boldsymbol{\epsilon}_\theta(\bar{\boldsymbol{z}}_t, t, C)$ and $\boldsymbol{\epsilon}_\theta(\bar{\boldsymbol{z}}_t, t, \varnothing_t)$ are equal. $\qquad\square$

Since we initialize $\bar{\boldsymbol{z}}_T = \boldsymbol{z}_T^*$ at diffusion step $T$, recursive application of Proposition 2 shows, under the ideal situation that the model has learned perfectly, that one will have $\bar{\boldsymbol{z}}_t = \boldsymbol{z}_t^*$ for all $t$ via letting $\varnothing_t = C$. In other words, one can regard that null-text inversion optimizes the unconditional prediction to approach the conditional prediction at each diffusion step.

Under practical situations, one can no longer expect the equality $\boldsymbol{\epsilon}_\theta(\boldsymbol{z}_t^*, t, C) = \boldsymbol{\epsilon}_\theta(\boldsymbol{z}_{t-1}^*, t-1, C)$ to hold. One can still expect, however, that the above equality approximately holds: One typically takes small timesteps so that $\alpha_{t-1} \approx \alpha_t$ and $\boldsymbol{z}_{t-1}^* \approx \boldsymbol{z}_t^*$, so that the argument given after Proposition 1 assures that the above equality holds approximately.

## A.2 EMPIRICAL EVALUATIONS

The assumption of perfect learning of the model adopted in Proposition 2 in the previous section is certainly too strong to be applied to practical situations. We have already discussed the issue of conditioning on $\boldsymbol{z}_0$ in the previous section. Another reason is that it is almost always the case that the model learns only approximately. Accordingly, what one can expect in practice would be that $\bar{\boldsymbol{z}}_t = \boldsymbol{z}_t^*$ holds only approximately, which would then make the validity of the optimality of $\varnothing_t = C$ in null-text inversion rather questionable. In this section, we investigate empirically how good the prompt-text embedding $C$ is compared with the optimized null-text embedding $\varnothing_t$, in terms of the noise prediction by the model, as well as their representation in the embedding space. In the experiments in this section, we used the same 100 image-prompt pairs from the COCO dataset as those used in the experiments in the main text.

We first investigated how close the noise prediction $\boldsymbol{\epsilon}_\theta(\boldsymbol{z}_t, t, \varnothing_t)$ using the optimized null-text embedding $\varnothing_t$ and the prediction $\boldsymbol{\epsilon}_\theta(\boldsymbol{z}_t, t, C)$ using the prompt embedding $C$ are. More

specifically, we performed null-text inversion, starting from $z_T^*$ obtained via DDIM inversion using the embedding $C$, and with the resulting sequences $(\bar{z}_t)_{t\in\{1,...,T\}}$ and $(\varnothing_t)_{t\in\{1,...,T\}}$ we evaluated the $L_1$ distance between $\epsilon_\theta(\bar{z}_t, t, \varnothing_t)$ and $\epsilon_\theta(\bar{z}_t, t, C)$. For comparison, we also calculated the $L_1$ distance between $\epsilon_\theta(\bar{z}_t, t, \varnothing_t)$ and the noise prediction $\epsilon_\theta(\bar{z}_t, t, C')$ obtained using the embeddings $C'$ of the prompts associated with images other than the target image, as well as the $L_1$ distance between $\epsilon_\theta(\bar{z}_t, t, C)$ and $\epsilon_\theta(\bar{z}_t, t, C')$. Figure 6 left shows the mean $L_1$ distance of the predicted noises. The predicted noises using the optimized embeddings $(\varnothing_t)_{t\in\{1,...,T\}}$ were closer to those using $C$ than those using $C'$, with a smaller distance than the distance between the predicted noise using $C$ and that using $C'$. One observes that the distance between the noise predictions using $\varnothing_t$ and $C$ became larger as $t$ became smaller, which would be ascribed to the accumulation of optimization errors. One can also notice that the distance between the noise predictions using $(\varnothing_t)_{t\in\{1,...,T\}}$ and $C$ were larger than that between those using $C$ and $C'$ near $t = 0$. Noise predictions near $t = 0$, however, would have almost no impact on generated samples since they are added at very small scales. The results suggest that the predicted noise $\epsilon_\theta(\bar{z}_t, t, \varnothing_t)$ using the optimized embedding $\varnothing_t$ in null-text inversion can be well approximated by the noise prediction $\epsilon_\theta(\bar{z}_t, t, C)$ using the embedding $C$ of the input prompt in (6).

We next calculated the cosine similarity in the 768-dimensional embedding space between the embeddings $C$ for 100 prompts and optimized embeddings $(\varnothing_t)_{t\in\{1,...,T\}}$ for each image. For each embedding sequence we took its average along the length of the sequence, and we centered the resulting average 768-dimensional prompt embeddings by subtracting the mean of 25,014 prompt embeddings, which are all the prompts included in the COCO validation dataset, and took a mean of embeddings over all tokens included in each prompt as the prompt embedding. Figure 6 right shows the mean cosine similarity. As $t$ became smaller, the similarity between the optimized null-text embedding $(\varnothing_t)$ and the embedding $C$ of the given prompt became positive, whereas the similarity between $(\varnothing_t)$ and embeddings $C'$ of the prompts for images other than the target image, as well as that between $C$ and $C'$, remained around zero. (We postulate that the small negative values of the similarity between $C$ and $C'$ throughout the entire range of $t$ are due to the bias induced from the centering.) This suggests that, although the implicit "meaning" represented by the optimized null-text embedding was almost orthogonal to the "meanings" of those of randomly-chosen prompts, it was closer to the "meaning" represented by the input prompt embedding $C$ in the region distant from $t = T$, as can be observed by the larger values of similarity between the optimized null-text embedding and the embedding of the input prompt (Optimized vs Input prompt). In the region distant from $t = T$, except the region near $t = 0$, the model is thought to generate detailed information about the image, which should be crucial in obtaining a high-quality reconstruction, so that the higher values of similarity in this region would suggest that embeddings that would be good in the sense of yielding a good reconstruction are closer to the embedding $C$ of the target prompt. In the large-$t$ region, on the other hand, the optimized null-text embedding $(\varnothing_t)$ had small similarity with the embedding $C$ of the given prompt, which can be ascribed to the fact that the null-text optimization is initialized with the same null-text embedding $\varnothing$, and is performed from $t = T$ down to $t = 1$. Note that, in the large-$t$ region, the similarity values were around zero because early stopping in optimizing $\varnothing_t$ was effective and optimization barely progressed.

From these results, we can say that the optimized embedding $\varnothing_t$ becomes semantically similar to the input prompt embedding $C$ as the optimization progresses. Therefore, it has been confirmed that our inversion method approximates null-text inversion.

## B  IMPLEMENTATION DETAILS

In our experiments, for the null-text inversion, we used the same settings at 50 sampling steps as those in the implementation available on the GitHub page of Mokady et al. (2023). Optimization was performed with the Adam optimizer, and the learning rate was set to reach $5 \times 10^{-3}$ at the last sampling step, changing linearly by the factor of $10^{-4}$ with the number of sampling steps. We further employed early stopping, and the threshold for early stopping was increased linearly in the number of sampling steps from $10^{-5}$ by the factor of $2 \times 10^{-5}$. We observed that when scheduling the learning rate and threshold with a function

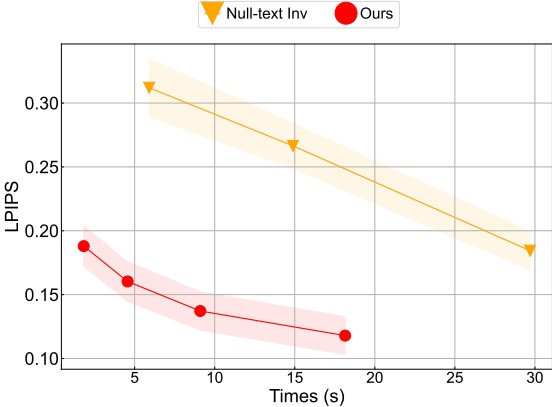

Figure 7: Comparison of LPIPS when calculation time is limited to less than 30 seconds.

of diffusion steps, the reconstruction quality was getting worse. See our code included in SM for details for more detailed implementation settings of our experiments.

## C  ADDITIONAL EXPERIMENTAL RESULTS

### C.1  COMPARISON OF RECONSTRUCTED IMAGES

Figure 7 shows a comparison of LPIPS between null-text inversion and our method when the computation time is limited to less than 30 seconds. The number of sampling steps in null-text inversion is 2, 5, and 10. LPIPS of null-text inversion below 10 sampling steps is degraded, and our method outperforms it. To consider realistic processing times, the reconstruction quality of our method is better than that of null-text inversion.

Figure 8 shows additional images reconstructed by the three methods compared. All the results show that DDIM inversion produced reconstructions that were not similar to the input images, while null-text inversion almost perfectly reconstructed the input images, and that our method also yielded results which were close to the reconstructions by null-text inversion.

### C.2  COMPARISON OF EDITED IMAGES USING PROMPT-OF-PROMPT

Figure 9 shows additional images edited by prompt-to-prompt. As can be seen, DDIM inversion failed to perform editing while maintaining the details of the original images. On the other hand, null-text inversion and the proposed method are both capable of editing while maintaining details of the original images, including object replacement and style changes.

### C.3  COMPARISON OF EDITED IMAGES USING OTHER EDITING METHODS

We demonstrate the advantage of the proposed method that it can be combined with various editing methods. For this purpose, we performed editing experiments by combining the proposal with other editing methods, SDEdit (Meng et al., 2022) and Plug-and-Play (Tumanyan et al., 2023). In SDEdit, a certain ratio $t_0$ is used as a hyperparameter to add noise to the sample $z_0$, and the latent variable $z_t$ at the diffusion step $t = t_0 \cdot T$ is obtained, which is then reconstructed by tracing the inverse diffusion process. For image editing, $z_0$ is obtained from the original image and an edited prompt is used during the inverse diffusion process calculation. We set the noisy sample $z_t$ calculated by DDIM inversion for null-text inversion and our negative-prompt inversion since they assume starting the sampling from $z_T$ calculated by DDIM inversion. In Plug-and-Play, the null-text is used as a prompt for

| Input | DDIM Inv | Null-text Inv | Ours |
|---|---|---|---|

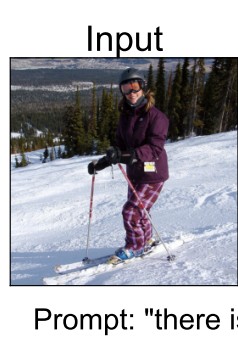 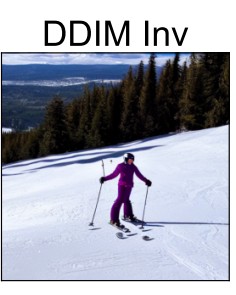 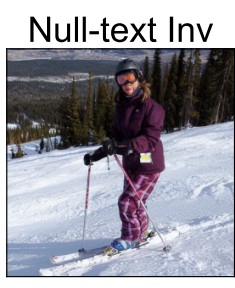 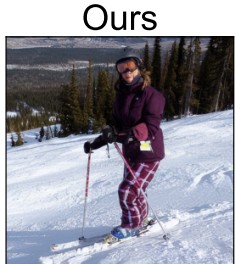

Prompt: "there is a woman about to ski down a hill"

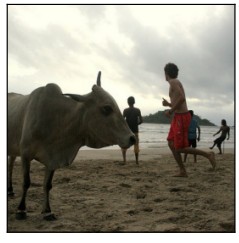 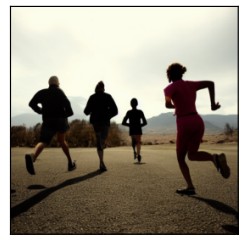 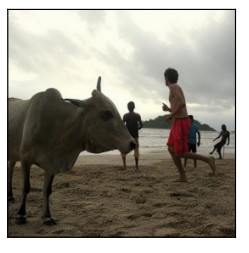 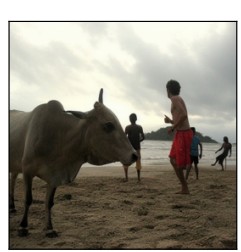

Prompt: "a few people running around some kind of animal"

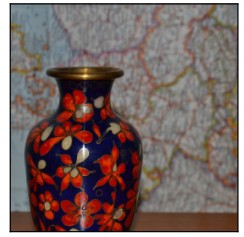 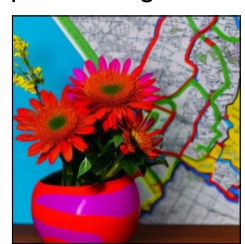 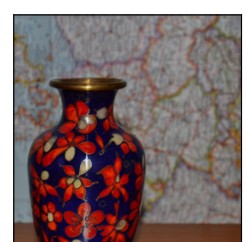 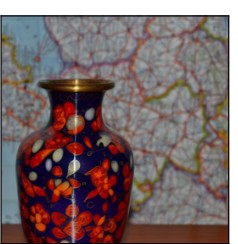

Prompt: "A brightly colored flower vase sits in front of a wall map."

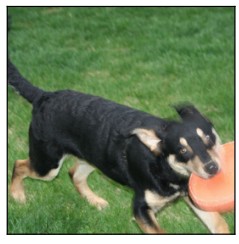 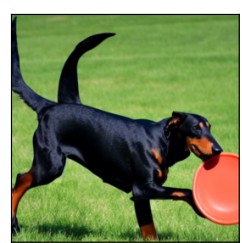 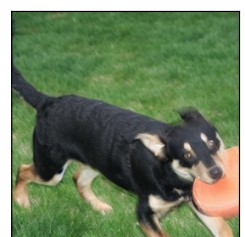 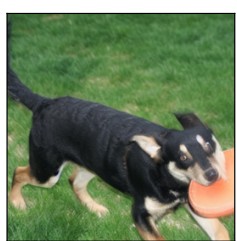

Prompt: "A black and brown doberman carries a frisbee."

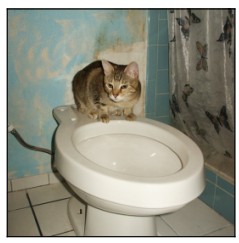 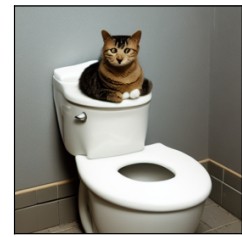 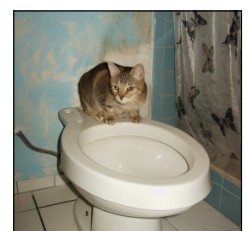 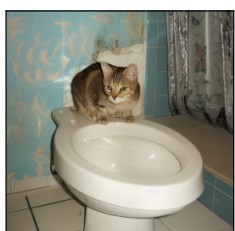

Prompt: "A cat sits on top of a toilet bowl."

Figure 8: Additional results of reconstructed images by the three methods.

Input          DDIM Inv     Null-text Inv          Ours

Prompt: "A yellow bird stands perched on a tree branch."

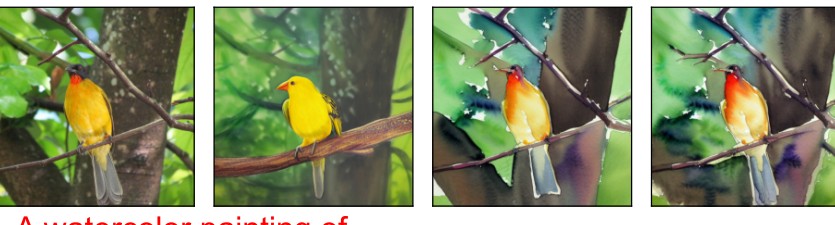

A watercolor painting of   ~

Prompt: "a few people running around some kind of animal"

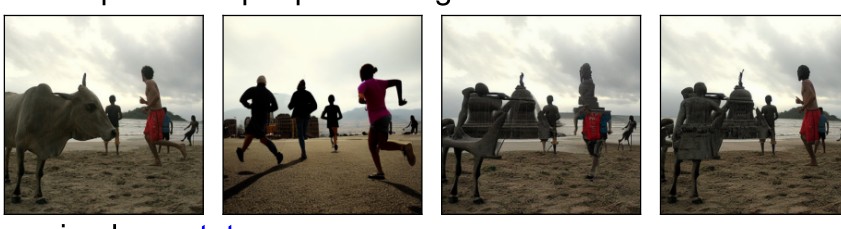

animal  →  statue

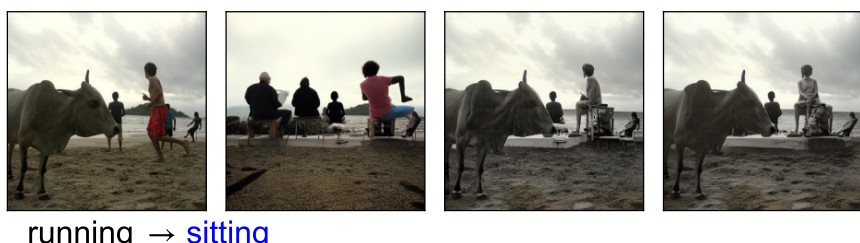

running  →  sitting

Prompt: "A black and brown doberman carries a frisbee."

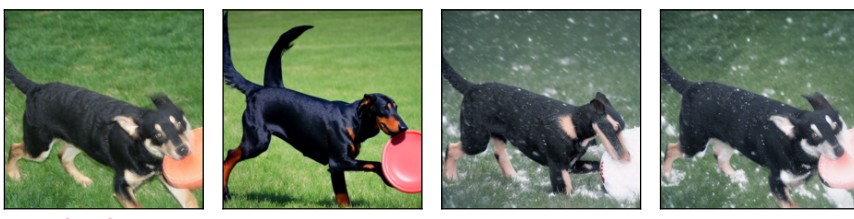

~   in the snow

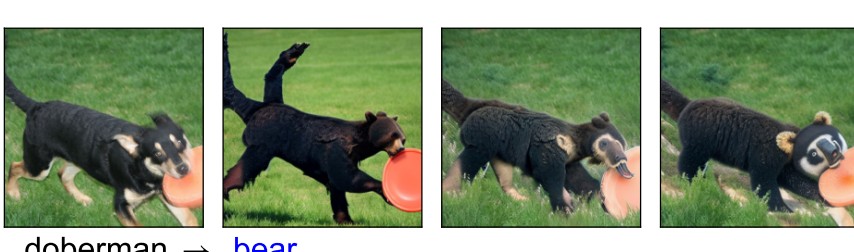

doberman  →  bear

Figure 9: Additional results of edited images by prompt-to-prompt combined with the three methods.

DDIM inversion. To combine the proposed method with it, we employ a prompt for the original image instead of the null-text for DDIM inversion.

Figure 10 shows images edited by SDEdit. As can be observed, SDEdit could not reconstruct the input images, while negative-prompt inversion and the proposed method were able to reconstruct details of the input images and appropriately edit them as specified by the prompts. Next, Figure 11 shows images edited by Plug-and-Play. Although the results of the proposed method are not generally better, the first, second, and fourth rows show better reconstruction quality and editing results in combination with the proposed method than the original method.

## C.4  FAILURE CASES

Figure 12 shows failure cases of our method. In all the cases shown, our method failed to reconstruct the images in 50 sampling steps, whereas null-text inversion successfully reconstructed them. The first two rows show failures due to the disappearance of people, where the objects were either reconstructed as non-human or as different persons. The third and fourth rows show failures due to the color gradient being reconstructed as separate objects, such as a single duck being reconstructed as scattered pieces, and a tree trunk being reconstructed as a different object. The last row shows a failure due to the disappearance of a tiny object, where one of the ski poles was missing. In the duck example, the reconstruction quality improved by increasing the number of sampling steps.

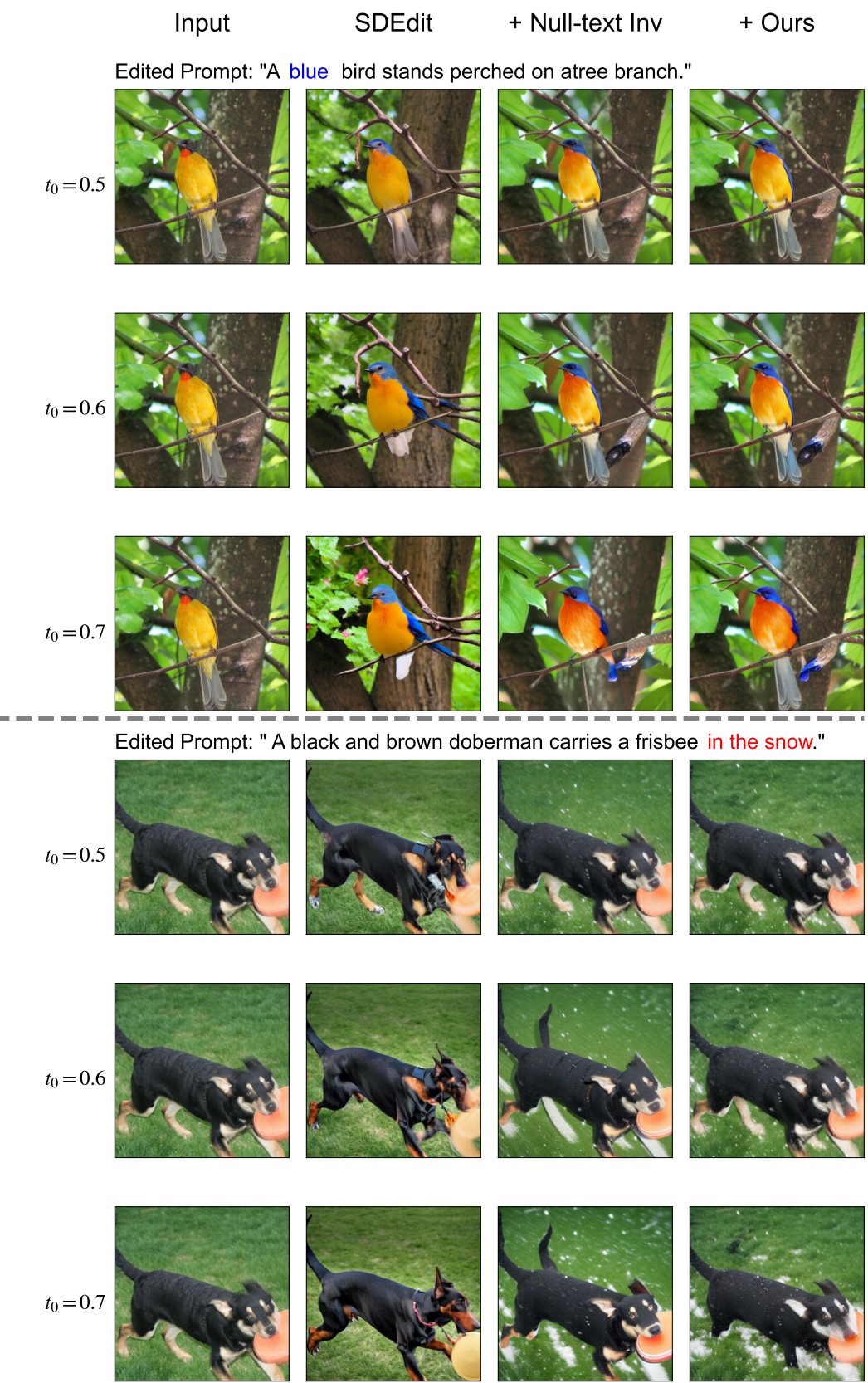

Figure 10: Additional results of edited images by SDEdit combined with null-text inversion or the proposed method.

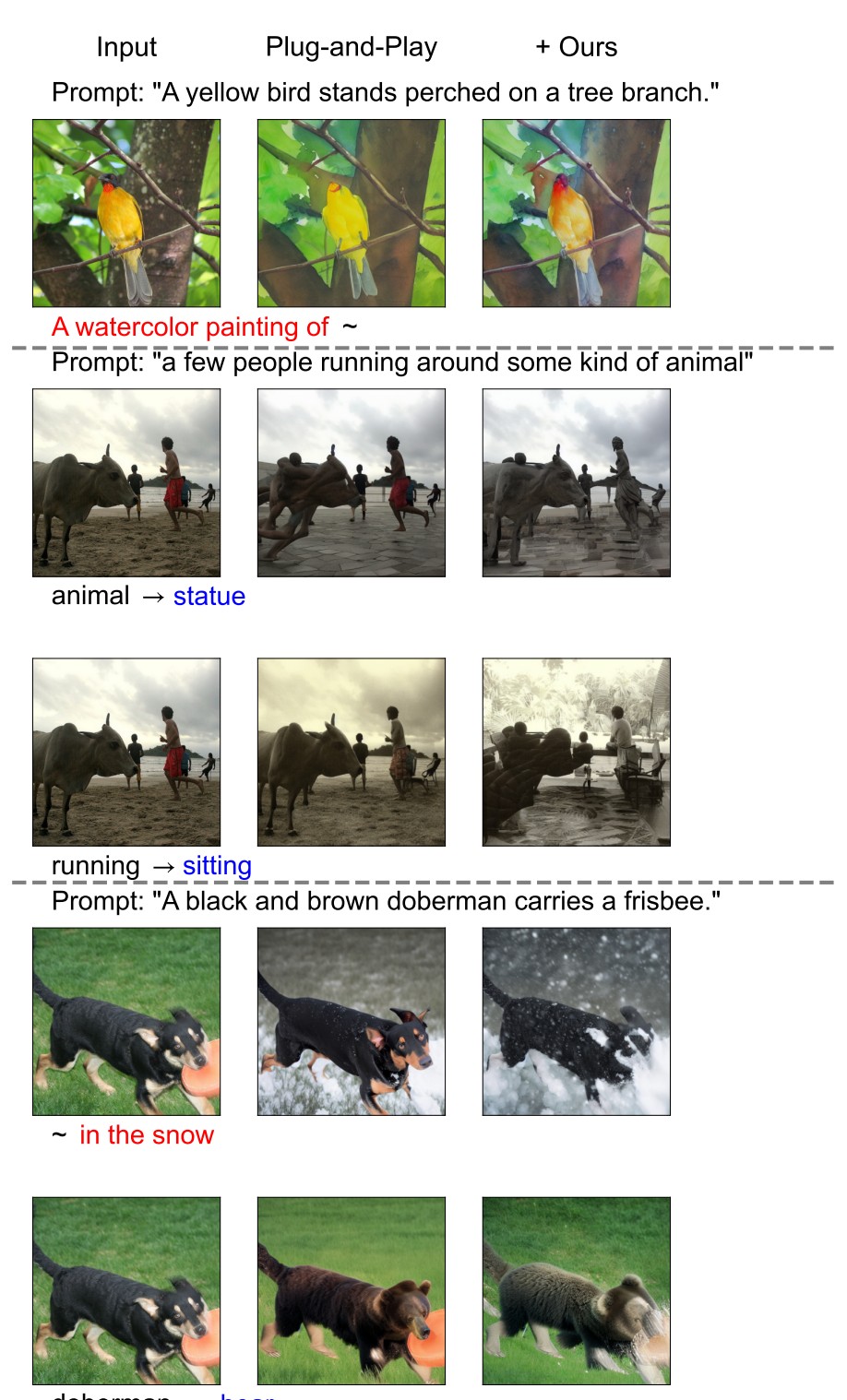

Figure 11: Additional results of edited images by Plug-and-Play combined with the proposed method.

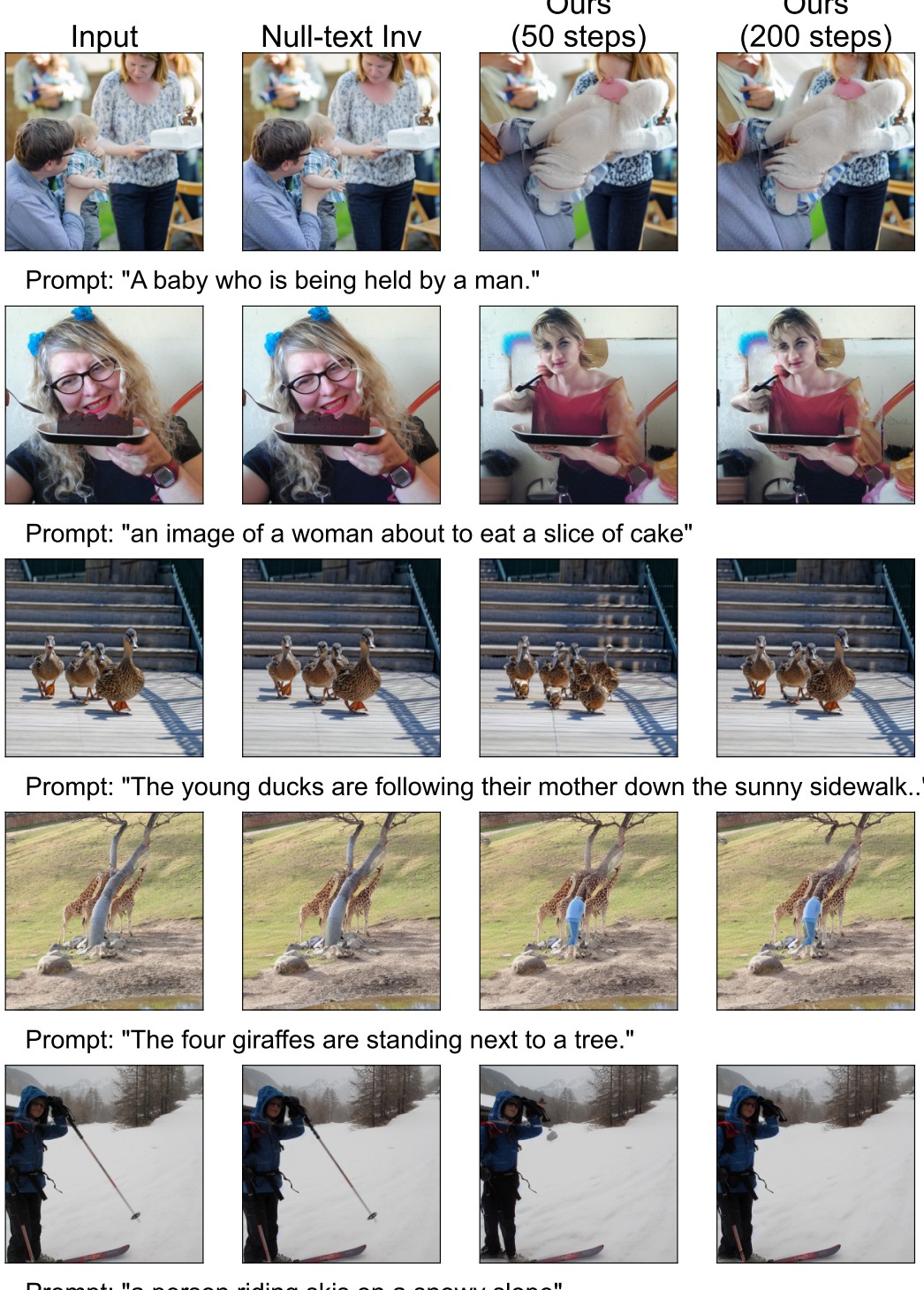

Figure 12: Additional failure cases of our method.

