# OpenReview forum: "Negative-prompt Inversion: Fast Image Inversion for Editing with Text-guided Diffusion Models"
_ICLR.cc/2024/Conference — Submitted to ICLR 2024_

### Official Review · Reviewer_69er · 2023-10-20

**Soundness:** 2 fair
**Presentation:** 2 fair
**Contribution:** 3 good
**Rating:** 6
**Confidence:** 4

**Summary:**

This paper proposes an efficient inversion technique for text-conditioned diffusion models. The aim of inversion is to recover the original image as faithfully as possible in the reverse process from the noisy latents obtained in the forward process of the diffusion model. Authors build on an existing technique that rely on solving an optimization problem for each time step in the reverse process, and propose a modification that does away with the costly optimization. The resulting fast inversion technique can be used with diffusion-based editing framework to perform purely text-based image editing.

**Strengths:**

- To the best of my knowledge, the idea to replace the null-text embedding with the fixed-prompt embedding is original.
- Speeding up realistic image editing via diffusion models has significant impact, as the compute cost of such models is the main factor hindering wider adoption.
- Based on the presented experiments, the speed-up can be significant (factor of 30x) while maintaining similar reconstruction quality to null-text inversion.

**Weaknesses:**

- I find the core assumptions of the paper difficult to understand and I think more justification/verification is needed (see questions below).
- The experiments could be more thorough, especially on editing. How does the performance of DDIM inversion scale with $w$? If I understand correctly it is only shown for $w=1$. Moreover, editing performance should be evaluated on the same benchmark as Mokady et. al (2023) (Table 2. in Mokady et. al).

**Questions:**

- Why would the predicted noise in adjacent steps equal (Eq. 6)? This would completely undermine the idea of iterative noising/denoising and would be very inaccurate for fairly large steps (such as N=50). How is the approximation impacted by the number of steps?

- If we approximate the optimized null-text embedding with C, it means that the same solution could have been found by null-text inversion, especially with shared embedding across time steps. How does null-text inversion compare with shared embeddings? Can we improve upon simply plugging in C by performing *some* optimization around C (not necessarily in every time step to reduce cost)?

- The experiments would be better presented in a way that compares inversion/editing performance *given a fixed time budget* in order to highlight the efficiency of the algorithm.

- I don't quite understand why the method is called negative-prompt inversion. Negative prompting commonly refers to a description of features in an image we do not want to generate in the context of text-conditioned diffusion models.

**Details Of Ethics Concerns:**

The proposed algorithm has the potential to enable fast and realistic image editing that could be used to produce misinformation or deep fakes.

---

> ### Author Response · Authors · 2023-11-22
>
> ## W1, Q1
> We have revised the justification regarding the assumption in Appendix A.1. In the revised version, we have argued after Proposition 2, that the “velocity field”, which the model is supposed to learn, is continuous in $z$ and $t>0$. It implies that, even though the equality $ \epsilon_{\theta}(z_t,t,C)=\epsilon_{\theta}(z_{t-1},t-1,C) $ does not hold exactly, one can expect that it holds approximately once one assume $ |\alpha_t - \alpha_{t-1}| $ is small.
>
> The reviewer’s concern that the accuracy of the approximation will worsen as the sampling step width increases is reflected in the experimental results in figure 4 left, which show that taking a smaller sampling step width improves the quality of reconstruction.
>
> ## W2
> We appreciate the reviewer for the thoughtful proposal. As the reviewer assumed, we have shown the results of DDIM inversion ($w=1$) followed by DDIM sampling ($w=7.5$) in the manuscript. One can also consider a method of DDIM inversion ($w=7.5$) followed by DDIM sampling ($w=7.5$), but this led to very bad results.
>
> Regarding the proposal to evaluate on the same benchmark as Mokady et al. (2023), we did not conduct that comparison since we could not identify the setting of editing and could not replicate the results of Table 2 in Mokady et al. (2023).
>
> ## Q2
> We appreciate the reviewer for the thoughtful question. As the reviewer suggested, we tried two experiments; One is to optimize a single embedding in null-text inversion, and the other is to initialize the optimized embedding by the prompt embedding $C$. In the former setting, we tried the same method as presented in section F of supplementary materials for Mokady et al. (2023), which is called global null-text inversion. Since we could see that this method could achieve the almost equal reconstruction quality to null-text inversion by taking about 20 times longer than the original, we calculated the cosine similarity of the optimized embedding and the prompt embedding in the same manner as in Appendix A.2. Since we could experiment with only some of the data due to time constraints, although we cannot exactly compare the results, the mean cosine similarity was $0.10$, and this result shows the optimized shared embedding was more similar to the prompt embedding than the optimized embedding in null-text inversion whose similarity have been shown in Appendix A.2, figure 6. We believe that longer optimization will make the optimized embedding more similar to the prompt embedding, however, getting too close to the prompt embedding will degrade the reconstruction quality. In the later setting, it yielded slight improvements in the reconstruction quality but did not beat null-text inversion. We are planning to continue analyzing why this setting cannot beat null-text inversion.
>
> ## Q3
> We appreciate the reviewer for this advice. We have added comparison of LPIPS when the computation time is limited to less than 30 seconds in Appendix C.1, figure 7. Considering realistic waiting time, our method outperformed null-text inversion in the reconstruction quality.
>
> ## Q4
> In text-guided image generation, if a negative-prompt is given, it is input to the model instead of the null-text in CFG calculation. As a result, the generated image diverges from the negative-prompt. Since our inversion takes a similar process in which uses a text prompt instead of the null-text, we named it “negative-prompt inversion”.

---

### Official Review · Reviewer_RErT · 2023-11-01

**Soundness:** 1 poor
**Presentation:** 2 fair
**Contribution:** 2 fair
**Rating:** 3
**Confidence:** 4

**Summary:**

This paper introduces negative-prompt inversion, a method that is capable of achieving comparable yet slightly degraded reconstruction quality as null-text inversion solely through forward propagation without optimization, thereby enabling much faster editing processes. Such an inversion technique is implemented by replacing the unconditional null-text prompt embedding with the conditional prompt embedding to modify the classifier-free guidance (CFG) in null-text inversion. Experiments demonstrate that the proposed negative-prompt inversion obtains comparable reconstruction quality as existing methods and is more than 30 times faster than null-text inversion. The authors also show that increasing sampling steps can further boost the reconstruction quality. Combining the proposed method with existing image editing methods like prompt-to-prompt allows fast real image editing.

**Strengths:**

- The paper is generally easy to follow. The symbols, terms, and concepts are adequately defined.

- The proposed method is very simple and easy to understand. Sufficient details are provided.

- The relevant literature is well-discussed and organized.

**Weaknesses:**

- The reviewer's primary concern is the actual soundness of the proposed negative-prompt inversion. To the reviewer's understanding, replacing the unconditional null-text embedding with the conditional prompt embedding in null-text inversion is akin to/the same as DDIM inversion without CFG. It is necessary to provide the results of DDIM inversion w/o CFG and compare it with the proposed negative-prompt inversion to verify its soundness. If the proposed negative-prompt inversion has the same effect as DDIM inversion without CFG, such contribution is a bit slim.

- More detailed discussions and analyses on the computational cost and memory usage should be provided since the authors claim them as one of the main contributions.

- It is advisable to avoid too many detailed discussions on the relevant studies in the Introduction section, which can be moved to the Related Work section. Also, the general idea of the proposed method should be briefly discussed in the Introduction. The previous version is a bit vague.

- The Method section also presents too many preliminaries and background on DDIM inversion, CFG, and null-text inversion. Such content can be shortened since the information is generally well-known.

**Questions:**

- Will the authors release all the code, models, and data to verify the soundness and ensure the reproducibility of this work?

- Line 2 of Abstract: Image editing not only changes the style of the image. Sometimes it involves certain semantics or geometry changes.

- The authors mentioned that by parallelizing and optimizing the program, there is potential to further accelerate their, where even real-time processing would be possible. The reviewer is interested in how to parallelize the program since reverse diffusion is an iterative process.

- Section 7 on Page 10: Please remove any main paper content beyond Page 9 to avoid the potential of template/formatting violations of the conference.

---

> ### Author Response · Authors · 2023-11-22
>
> ## W1
> We appreciate the reviewer's insightful comments regarding the comparability of our method to DDIM inversion without CFG, specifically in the context of reconstruction. We acknowledge the similarities highlighted; however, our approach distinctly focuses on editing by employing a source prompt instead of the null-text, which marks a significant difference from the standard DDIM framework. To elucidate this distinction more clearly, we have revised the itemization in Section 3.4 on page 6.
>
> ## W2
> We thank the reviewer for this comment. The term of computational cost represents the computational time, whose results have been shown in figure 4. In memory usage, our method uses half the memory of null-text inversion. We have added this description at the end of Section 4.2.
>
> ## W3
> We appropriate the reviewer for this comment. We have removed expressions that were considered redundant and clarified the general idea of our method in the third paragraph of the Introduction section.
>
> ## W4
> We have trimmed and shortened the explanations that we considered unnecessary in the Method section.
>
> ## Q1
> We have included our code in supplementary materials for the evaluation purpose by the reviewers, as described in the reproducibility statement of the manuscript. Additionally, we will publicly release our code if our paper is accepted.
>
> ## Q2
> We thank the reviewer for pointing this out. According to the author guide, we cannot change the abstract except in the camera-ready version, so we are planning to change the corresponding part, “changing the style” to “changing the subject and the style”, if our paper is accepted.
>
> ## Q3
> We thank the reviewer for this question. We have revised  the description regarding parallelization in the conclusion section. We assumed parallelization to mean processing several frames simultaneously using multiple GPUs, which improves not the latency but the throughput, not parallelizing the reverse diffusion process which is iterative as the reviewer assumed.
>
> ## Q4
> In response to the reviewer’s comment, we have revisited the author guide for ICLR 2024, which explicitly states that the reproducibility statement should be placed at the end of the main text and before the reference list, and that it will not count toward the page limit. We thus believe that our manuscript does conform to the formatting instructions.

---

### Official Review · Reviewer_eMct · 2023-11-02

**Soundness:** 3 good
**Presentation:** 3 good
**Contribution:** 1 poor
**Rating:** 5
**Confidence:** 4

**Summary:**

The paper introduces the Negative Prompt Inversion method, which is designed for fast image reconstruction in the context of diffusion models, with a particular focus on image editing. The primary motivation behind this work is to achieve high-quality image reconstruction while reducing the computational cost and processing time involved.

The paper builds upon null-text inversion, a technique that leverages the denoising diffusion implicit model (DDIM) inversion and Classifier Free Guidance (CFG). Null-text inversion optimizes the embedding vector of an empty string to align the diffusion process calculated by DDIM inversion with the reverse diffusion process calculated using CFG. Negative Prompt Inversion is based on the observation that using the origin (resp. target) prompt instead of optimizing the null prompt brings comparable results in reconstruction (resp. edition). This simple modification offers a substantial improvement in processing speed.

**Strengths:**

- **Well-written paper**

- **Technically sound** The paper provides a clear summary of preliminary works as well as extensive justifications.

**Weaknesses:**

- **Lack of comparison with existing baselines** The paper could benefit from a more comprehensive comparison with existing baseline methods. Specifically, it does not compare the proposed Negative Prompt Inversion with recent image editing methods based on prompt interpolation, such as Imagic or UniTune. This comparison would help assess the relative strengths and weaknesses of the proposed method in the context of image editing. Moreover, the proposed work as well as the Null-text inversion are also very close to the prompt tuning inversion paper and could also be compared to it.

- **Mixed performances** The paper mentions that the image editing performances of Negative Prompt Inversion are below DDIM with CFG. This observation raises questions about the practical utility of the proposed method.

- **Lack of justification** The paper does not sufficiently justify the underlying hypotheses and assumptions of the Negative Prompt Inversion method in the main paper. Specifically, the paper relies on two strong hypotheses regarding the equivalence of null-text inversion features and initial DDIM trajectories, as well as the equality of predicted noise at adjacent denoising steps. These hypotheses are not adequately explained or justified in the main paper and are relegated to the supplementary material. Providing a more robust rationale for these assumptions would enhance the paper's credibility and comprehensibility. Furthermore,  Proposition 3 in the supplementary is supposed to justify the second hypothesis in the general case, but it lacks clarity. In particular, the implication that $\alpha_t \simeq \alpha_{t-1} \Rightarrow z_t^{\ast} \simeq z_{t-1}^{\ast}$ should be clarified and expanded.

**Questions:**

- Are the edited images provided in the supplementary Fig. 8, second column, obtained with DDIM + CFG or just DDIM? In the former case, the results seem in disagreement with the CLIP score shown in Table 1. In the latter case, editing results with DDIM + CFG could be provided here.

- Most of the justifications in the supplementary material (sec. A.1)  are provided for the case where the noise component is conditioned on the clean sample $z_0$. In this case, the conclusion holds $\bar{z}_t = z^*_t$. However, as stated later, the denoising model does not have $z_0$ as input. If it does not support the justification in the actual process of DDIM sampling, I wonder how useful are Prop. 1 and all the discussion from eq. (7) to (23)?

---

> ### Author Response · Authors · 2023-11-22
>
> ## W1
> We thank the reviewer for this comment. As the reviewer suggested, we have added results of Imagic for comparison, which shows that our method also outperformed Imagic. We also tried UniTune and prompt tuning inversion, but we failed to replicate the respective papers’ results since the official codes are not released.
>
> ## W2
> CLIP score calculates the similarity between an image and its prompt, and it does not regard how an edited image is similar to the original one. By summarizing the evaluation both via LPIPS and CLIP score, we believe that the editing performance of our method is not below DDIM with CFG.
>
> ## W3
> We thank the reviewer for this comment. We have revised the justification regarding the assumption in Appendix A.1.
> The assumption of equivalence of null-text inversion features and DDIM inversion trajectory is just for the mathematical induction: One has $ z^{*}_T = \bar{z}_T $, and we argue that, assuming that the equivalence holds true at diffusion step $ t $, it should hold true at diffusion step $ t-1 $.
>
> The second hypothesis is related to Q2, so we have answered it in Q2.
>
> Regarding the statement $ \alpha_t \approx \alpha_{t-1} \Rightarrow z_t^* \approx z_{t-1}^* $,which the reviewer pointed out, it is justified by (3), since $ z_t^* $ is calculated by DDIM inversion. Notice that in (3) one has $ z_{t+1} \rightarrow z_t $ as $ \alpha_{t+1}  \rightarrow \alpha_t $.
>
> ## Q1
> As the reviewer pointed out, it was difficult to tell what the description of “DDIM Inv” represents. The description “DDIM Inv” represents DDIM inversion followed by DDIM + CFG. We have aligned the description in the manuscript. Regarding the disagreement with the CLIP score, we think it as follows: It is true that DDIM inversion fails to generate snow in the fourth row of figure 9, but the other descriptions for a Doberman are correct, which could be leading not to low the CLIP score. In fact, the CLIP score of the image is $27.25$, which is higher than the mean score.
>
> ## Q2
> We thank the reviewer for this question. We have extensively rewritten Appendix A.1. In the revised version, we have argued after Proposition 2, that, even when one considers the sample distribution rather than a single sample, the “velocity field”, which the model is supposed to learn, is continuous in $z$ and $t>0$. It implies that, even though the equality $ \epsilon_\theta(z_t^*,t,C)=\epsilon_\theta(z_{t-1}^*,t-1,C) $ does not hold exactly, one can expect that it holds approximately once one assumes $ |\alpha_t-\alpha_{t-1}| $ is small.

---

### Official Review · Reviewer_8NbU · 2023-11-06

**Soundness:** 3 good
**Presentation:** 4 excellent
**Contribution:** 3 good
**Rating:** 5
**Confidence:** 4

**Summary:**

This paper proposed a method called negative prompt inversion that modified null text inversion for diffusion models and achieved a faster inversion method with forward computation only without optimizing, i.e., optimizing the null text embedding.  More specifically, in conventional null text inversion, one can optimize the null text embedding so that the predicted $z_{t-1}$ with CFG and $z_{t-1}^*$ in DDIM inversion. In this paper, instead, the authors investigated the conditions/constraints on noise $\epsilon_{\theta}$ to make $z_{t-1}$ equal to  $z_{t-1}^*$, which leads to a simper and faster inversion method. The methods are evaluated on 100 randomly selected COCO images with quality metrics like PSNR, LPIPS, etc., and speed.

**Strengths:**

1) The proposed idea of negative prompt inversion that only needs forward computation  but not optimization is qutie interesting.

2) The paper is organized and presented quite well, and friendly to understand, and easy to follow. Moreover, the introduction and related works parts are also quite helpful and informative to provide the big picture and the motivation.

3) Some promising results are shown in the experiments, with 30 times faster inversion than null text inversion methods.

**Weaknesses:**

1) The assumption that the predicted noises at adjacent diffusion steps are equal seems neither rigorous nor practical. The authors may need to provide more justification why this assumption is valid.

2) The evaluation is done with only 100 COCO images, which is quite small. Moreover, only objective metrics are provided. Human subjective evaluation should also be provided since it is more reliable to judge the quality, which is quite easy to do considering the data set is small.

3) The authors claimed some limitations about can not reconstruct faces well. It will be helpful to provide some failure face cases (and failure cases beyond faces if there are).

**Questions:**

I actually like the proposed idea and this paper overall. If authors can resolver the questions in the weakness section, I will be happy to increase my rating.

**Details Of Ethics Concerns:**

This paper is about a new image inversion/editing method, which may have harmful applications.

---

> ### Author Response · Authors · 2023-11-22
>
> ## W1
> We thank the reviewer for this comment. The assumption on the predicted noises at adjacent diffusion steps, which the reviewer pointed out, has been argued in Appendix A.1. The model is supposed to learn the “velocity field”, which is shown in Proposition 1 in the revised manuscript to be continuous in $z$ and $t$. So the assumption that the predictions at adjacent diffusion steps are approximately equal holds provided that one takes small diffusion steps. Additionally, we have revised the justification regarding the assumption in Appendix A.1.
> ## W2
> We appreciate the reviewer for this comment. We evaluated our method by using 100 images in the same manner as in the original paper of Null-text Inversion (Mokady et al. CVPR2023).
> Regarding human subjective evaluation, we did not conduct it since it was difficult for us to prepare it and to collect dozens of subjects during the short discussion period. We think that several editing images in the manuscript including those in appendices should be helpful for judging our editing quality.
> ## W3
> We thank the reviewer for this comment. We have provided some failure cases, including an instance of face reconstruction, in Appendix C.4, figure 12. Please refer to this section.

---

### Author Response · Authors · 2023-11-22

We thank the reviewers for their thoughtful reviews. We have revised the manuscript and added, after the main text, the appendices, which were provided as the supplementary materials in the initial submission.

The revisions are as follows:
 - In the third paragraph of the Introduction section, we have removed the detailed explanation of CFG and DDIM inversion, and we have added a simple explanation of how to edit images with our method.
 - In Section 3.2 DDIM inversion and Section 3.3 Null-text inversion, we have removed expressions that were considered redundant.
 - In Section 3.4 Negative-prompt inversion, we have revised to make it easy to understand the difference of reconstruction and editing by our method.
 - At the end of Section 4.2 Reconstruction, we have added a description of memory usage comparison in order to demonstrate that our method uses less memory.
 - In the second paragraph of Section 4.3 Editing and Table 1, we have added the results of Imagic as another image editing method to compare them with those of our method.
 - In the second paragraph of the Conclusions section, we have added a more detailed explanation of how to further accelerate our method.
 - Appendix A.1 has been extensively revised. In particular, we have clarified that the “velocity field”, which the model is supposed to learn, is continuous in z and t>0 even when one considers the sample distribution rather than a single sample. We believe that this modification makes our justifying argument clearer.
 - In Appendix C.1, we have added comparison by our method and null-text inversion when calculation time is limited in order to more highlight the fastness of our method.

We have indicated the revised parts by coloring them in blue in the revised version.

---

### Meta-Review · Area_Chair_aiNq · 2023-12-05

**Metareview:**

In this paper, authors proposed negative-prompt inversion, a method to reconstruct original images with diffusion model through forward propagation without optimization. Authors showed that their methods is more than 30 times faster than null-text conversion. Combining their method with existing image editing methods like prompt-to-prompt allows fast image editing. The strengths and weaknesses given by reviewers are. Strengths: 1) the proposed method only needs forward computation but not optimizing is quite interesting; 2) paper is easy to follow; 3) results promising. Weaknesses are: 1) lack of experimental result and ablation studies to justify the contribution of this work; 2) quality is actually worse than DDIM with CFG; 3) some technique details are missing made reviewers hesitate to give higher ratings.

Before rebuttal, reviewers' score are: 2 "5: marginally below the acceptance threshold", 1 "3: reject, not good enough", 1 " 6: marginally above the acceptance threshold". No additional reviewers' feedback is given after rebuttal.

**Justification For Why Not Higher Score:**

Missing additional experimental results and technique details made reviewers hesitate to fully support the acceptance of this paper.

**Justification For Why Not Lower Score:**

NA

---

### Decision · Program_Chairs · 2024-01-16

Reject